

# Estimation of ground level particulate matter concentrations through the synergistic use of satellite observations and process-based models over South Korea

Seohui Park[1,*] Minso Shin[1,*], Jungho Im[1], Chang-Keun Song[1], Myungje Choi[2], Jhoon Kim[2], Seungun Lee[3], Rokjin Park[3], Jiyoung Kim[4], Dong-Won Lee[5], and Sang-Kyun Kim[5]

[1]School of Urban & Environmental Engineering, Ulsan National Institute of Science and Technology, Ulsan, 44919, Republic of Korea
[2]Department of Atmospheric Sciences, Yonsei University, Seoul, 03722, Republic of Korea
[3]School of Earth and Environmental Sciences, Seoul National University, Seoul, 08826, Republic of Korea
[4]Global Environment Research Division, Climate and Air Quality Research Department, National Institute of Environmental Research, Incheon, 22689, Republic of Korea
[5]Environmental Satellite Centre, Climate and Air Quality Research Department, National Institute of Environmental Research, Incheon, 22689, Republic of Korea

[*]These authors equally contributed.

*Correspondence to*: Jungho Im (ersgis@unist.ac.kr)

**Abstract.** The long exposure to particulate matter (PM) with aerodynamic diameters < 10 μm ($PM_{10}$) and 2.5 μm ($PM_{2.5}$) has negative effects on human health. Although station-based PM monitoring has been conducted around the world, it is still challenging to provide spatially continuous PM information for vast areas at high spatial resolution. Satellite-derived aerosol information such as aerosol optical depth (AOD) has been frequently used to investigate ground-level PM concentrations. In this study, we combined multiple satellite-derived products including AOD with model-based meteorological parameters (i.e. dew-point temperature, wind speed, surface pressure, planetary boundary layer height, and relative humidity) and emission parameters (i.e. NO, $NH_3$, $SO_2$, POA, and HCHO) to estimate surface PM concentrations over South Korea. Random forest (RF) machine learning was used to estimate both $PM_{10}$ and $PM_{2.5}$ concentrations with a total of 32 parameters for 2015-2016. The results show that the RF-based models produced good performance resulting in $R^2$ values of 0.78 and 0.73, and RMSEs of 17.08 μg/m$^3$ and 8.25 μg/m$^3$ for $PM_{10}$ and $PM_{2.5}$, respectively. In particular, the proposed models successfully estimated high PM concentrations. AOD was identified as the most significant for estimating ground-level PM concentrations, followed by wind speed, solar radiation, and dew-point temperature. The use of aerosol information derived from a geostationary satellite sensor (i.e., GOCI) resulted in slightly higher accuracy for estimating PM concentrations than that from a polar-orbiting sensor system (i.e., MODIS). The proposed RF models yielded better performance, particularly in improving on the underestimation of the process-based models (i.e., GEOS-Chem and CMAQ).



## 1 Introduction

Epidemiological studies have consistently shown that negative human health effects including premature mortality can be caused by long-term exposure to atmospheric aerosols and particles, especially $PM_{10}$ and $PM_{2.5}$ (particulate matter with an aerodynamic diameter of less than 10 μm and 2.5 μm, respectively) (Pope III et al., 2009; Bartell et al., 2013; Jerrett et al.,

2017). Consequently, the monitoring and assessment of exposure to $PM_{10}$ and $PM_{2.5}$ are crucial for effective management of public health risks. In recent decades, East Asia has been significantly industrialized and urbanized through its rapid economic growth. The industrialization and urbanization have resulted in adverse effect on air quality not only in this region but also in neighbouring countries (Koo et al., 2012).

The Public Health and Environment Research Institute in South Korea has been monitoring $PM_{10}$ and $PM_{2.5}$ concentrations

at numerous sites all over its jurisdiction. Even though the distribution of the monitoring sites is relatively dense, there is a limitation in providing spatially continuous particulate matter (PM) concentrations that focus on major urban areas. For example, Zang et al. (2017) studied the effect of a temperature inversion layer on the relationship between aerosol optical depth (AOD) and $PM_{2.5}$. The aerosol robotic network (AERONET) AOD and radiosonde data were used to estimate ground $PM_{2.5}$ concentrations through an optimized subset regression model. They found the temperature inversion layer to be a key

factor in enhancing the accuracy of a ground-level $PM_{2.5}$ estimation model with a coefficient of determination ($R^2$) of 0.63 and a root mean square error (RMSE) of 35.45 μg/m$^3$ (Zang et al., 2017). Ground-based data typically have uncertainty for spatial distribution of PM concentrations as they are point-based measurements requiring spatial interpolation. On the other hand, satellite-based PM monitoring has the potential to provide information on air quality over vast areas at high spatial resolution. Many studies have examined the use of satellite-based products to estimate surface PM concentrations (Liu et al.,

2005; Gupta and Christopher, 2009a,b; Van Donkelaar et al., 2010, 2015; Chudnovsky et al., 2014; Li et al., 2015; Xu et al., 2015a; You et al., 2015; Wu et al., 2016). AOD is the most widely used parameter that can be derived from satellite remote sensing to estimate ground-level PM concentrations. It represents the amount of light attenuation caused by atmospheric aerosol scattering and absorption in the vertical column.

Early studies generally adopted simple linear regression to investigate the relationship between total column AOD and

surface PM concentrations (Liu et al., 2005; Liu et al., 2007). Liu et al. (2005) estimated ground-level $PM_{2.5}$ concentrations over the eastern United States using Multiangle Imaging Spectroradiometer (MISR)-derived AOD, Planetary Boundary Layer Height (PBLH) and Relative Humidity (RH) from the Goddard Earth Observing System (GEOS-3). Their results yielded an $R^2$ of 0.48 and an RMSE of 13.8 μg/m$^3$ when the estimated $PM_{2.5}$ concentrations were compared to in-situ measurements. More recent studies explored advanced statistical approaches to improve the prediction of ground-level PM

concentrations such as mixed-effects models, geographically weighted regression (GWR), support vector machines (SVM), and artificial neural networks (ANN), as well as the use of chemical transport models (CTM). Van Donkelaar et al. (2010) combined Moderate Resolution Imaging Spectroradiometer (MODIS) and MISR-derived AODs, and multiplied them to the ratio between $PM_{2.5}$ and AOD simulated by the GEOS-Chem model (i.e., CTM) to estimate global 6-year (2001-2006)



averaged PM$_{2.5}$ concentrations. Their results showed a strong spatial agreement with in-situ PM$_{2.5}$ concentrations in North America (slope = 1.07; R$^2$ = 0.59). To estimate daily PM$_{2.5}$ concentrations over the United States using random forest (RF), Hu et al. (2017b) incorporated MODIS AOD, simulated GEOS-Chem AOD, meteorological data, and land-use information. The developed RF model produced an R$^2$ of 0.8 and an RMSE of 2.83 μg/m$^3$ from 10-fold cross validation.

Most previous studies have mainly used AOD produced from polar orbiting satellite sensor systems such as MODIS and MISR. They provide AOD worldwide but only make it available once a day because of the revisit time. A major problem with daily AOD is cloud contamination. In particular, as the cloud cover rate is high in Asia, it is difficult to obtain spatially continuous AOD over the region. Therefore, many studies have focused on the United States to estimate ground-level PM concentrations using polar orbiting satellite data. AOD produced from geostationary satellite sensor systems may be a better

option for estimating ground level PM concentrations due to it having a higher temporal resolution than polar orbiting sensor systems. The Geostationary Ocean Colour Imager (GOCI) is the world's first geostationary ocean color satellite sensor that provides multi-spectral aerosol data in Northeast Asia (included eastern China, the Korea peninsula, and Japan) (Park et al., 2014; Xu et al., 2015a). GOCI provides hourly data at 500 m resolution 8 times a day from 9:00 to 16:00 Korean Standard time (KST). Xu et al. (2015a) examined PM$_{2.5}$ concentrations in eastern China using GOCI-derived AOD, coupled with

GEOS-Chem simulation data, resulting in a strong correlation (R$^2$ = 0.66) with in-situ measurements in terms of annual mean concentrations.

In addition, recent studies have used PBLH, RH, wind speed, and other meteorological variables and land use information because these factors are related to PM concentrations, and thus can be used to improve estimation models (Gupta and Christopher, 2009a; Liu et al., 2009; Wu et al., 2012; Chudnovsky et al., 2014; You et al., 2015; Wu et al., 2016; Li et al.,

2017b; Yeganeh et al., 2017). In this study, we adopted the machine learning approach, Random Forest (RF), to develop models estimating ground level PM$_{10}$ and PM$_{2.5}$ concentrations using satellite-derived products, numerical and emission model output, and ancillary spatial data over South Korea. Aerosol products retrieved from GOCI including AOD were used as key input variables. The objectives of this study are to (1) estimate ground-level PM$_{10}$ and PM$_{2.5}$ concentrations based on GOCI aerosol products and meteorological and emission model output data using RF, (2) validate the estimated PM

concentrations using in-situ observation data, (3) compare the results to those when MODIS aerosol products were used instead of GOCI products, (4) evaluate the proposed remote sensing-based models in comparison with the results from physical models such as GEOS-Chem and the Community Multiscale Air Quality Modelling System (CMAQ).

## 2 Study area and data

### 2.1 Study area

The study area was South Korea (latitude: 33°N-39°N, longitude: 124°E-131.5°E), located in northeast Asia, a region known to have relatively poor air quality. Our study area is located in the mid-latitude region where the prevailing westerlies carry particulates from the two most rapidly developing countries in Asia (i.e., China and India). The annual mean temperature of





South Korea ranges from 10 to 15°C, and the annual precipitation ranges from 1000 to 1900 mm. More than half of the precipitation occurs in summer during the Asian monsoon. Wind direction is seasonal, with north-westerly winds prevailing in winter and south-westerly winds in summer.

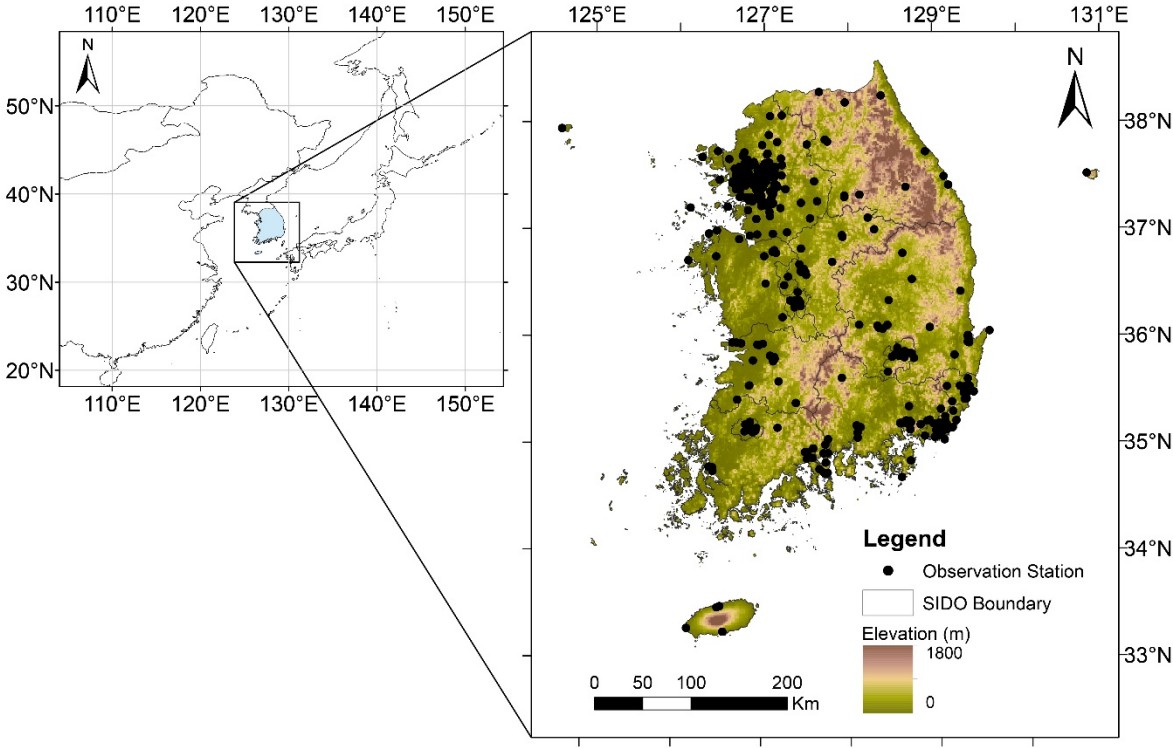

**Figure 1: Study area with particulate matter (PM) monitoring station sites in South Korea. Elevation is used as a background image.**

## 2.2 Data

### 2.2.1 Observation data

PM observation data (i.e. $PM_{10}$ and $PM_{2.5}$) in South Korea were obtained from the AirKorea website (https://www.airkorea.or.kr/) for the period from 2015 to 2016. A total of 325 stations are distributed throughout the country with a concentration in metropolitan areas such as the Seoul Metropolitan Area (SMA) (Figure 1). Hourly concentrations of air pollutants such as $PM_{10}$ and $PM_{2.5}$ are provided as real time data. Currently, $PM_{10}$ data are provided at 316 stations while $PM_{2.5}$ are measured at 194 stations.



### 2.2.2 Remote sensing data

GOCI is a geostationary satellite imaging sensor onboard the Communication, Ocean, and Meteorological Satellite (COMS), which was launched in June 2010. It covers 2500 km x 2500 km over the East Asia region and 8 images collected at 6 visible and 2 NIR bands per day provided hourly from 09:00 to 16:00 in local time (KST). GOCI aerosol products are derived by GOCI Yonsei aerosol retrieval (YAER) version 2 algorithm (Choi et al., 2018). Four types of products were used in this study: AOD at 550 nm, fine-mode fraction (FMF) at 550m, single scattering albedo (SSA) at 440 nm, and Ångström exponent (AE) between 440 and 870 nm with 6 km x 6 km of spatial resolution (Table 1).

MODIS, onboard the Terra and Aqua satellites, acquires data in 36 spectral bands ranging from 0.4 to 1.4 μm in wavelength, to observe the Earth's environment. The 16-days Normalized Difference Vegetation Index (NDVI) with 1 km resolution (MYD13A2; Solano et al., 2010) and Aerosol 5-min L2 swath data with 3km resolution (MYD04_3K; Levy et al., 2013) products from 2015 to 2016, and the yearly land cover type product with 500 m resolution (MCD12Q1; Friedl et al., 2010) in 2013 were obtained from Reverb Echo (https://reverb.echo.nasa.gov/reverb/). Urban area ratios were calculated using land cover data based on the 13 x 13 neighbourhood pixels, which were similar to the spatial resolution of GOCI AOD products. MODIS Aerosol product was used for comparison with GOCI AOD data.

Global Precipitation Measurement (GPM; Huffman et al., 2015) developed by the National Aeronautics and Space Administration (NASA) and the Japanese Aerospace Exploration Agency (JAXA) was launched in February 2014 to provide observations of rain and snow worldwide. Half-hourly precipitation data with 0.1-degree resolution (3IMERGHH) were obtained from Goddard Earth Science Data and Information Service Centre (GES DISC; https://mirador.gsfc.nasa.gov/). Half-hourly precipitation data were provided as precipitation rates with mm/hr and used to calculate 24-hour accumulated precipitation data for every hour.

The Shuttle Radar Topography Mission (SRTM; Farr et al., 2007) was launched as a payload on the STS-99 mission of the Space Shuttle Endeavour to generate a global digital elevation model (DEM) of the Earth. SRTM DEM data were acquired by using the radar interferometry based on the C-band Spaceborne Imaging Radar (SIR-C) and the X-band Synthetic Aperture Radar (X-SAR) hardware. The elevation data were provided at 1 arc-second (about 30 meters) and 3 arc-second (about 90 meters) of spatial resolution for global coverage from the U.S. Geological Survey (USGS) EarthExplorer website (https://earthexplorer.usgs.gov/). In this study, 3 arc-second data were used and resampled to the same resolution as the MODIS data with 1 km of spatial resolution (Table 1).

**Table 1: Remote sensing data used to develop models estimating ground-level particulate matter concentrations in this study.**

| Product | Spatial resolution | Temporal resolution | Variables | Description |
|---|---|---|---|---|



| | | | | |
|---|---|---|---|---|
| GOCI AOD_550nm | 6 km | 8/day | Aerosol Optical Depth (AOD) | The measure of the extinction of the solar radiation by aerosols (e.g., dust, haze, and sea salt) |
| GOCI FMF_550nm | 6 km | 8/day | Fine Mode Fraction (FMF) | The ratio of small size aerosols (radii between 0.1 and 0.25) to the total aerosols |
| GOCI SSA_440nm | 6 km | 8/day | Single Scattering Albedo (SSA) | The measure of the amount of aerosol light extinction due to scattering |
| GOCI AE_440_870nm | 6 km | 8/day | Ångström Exponent (AE) | The exponent related with particle size (The smaller the particles, the bigger the Ångström Exponent) |
| MODIS MYD13A2 | 1 km | 16 days | Normalized Difference Vegetation Index (NDVI) | The indicator denoting vegetation quantification |
| MODIS MCD12Q1 | 500 m | yearly | Land Cover Type (Urban area ratio) | The ratio of urban area to 6 km x 6 km neighbourhood of each pixel |
| GPM 3IMERGHH | 0.1° | 30 min | Precipitation | The 24-h accumulated precipitation produced using 30 minutes 3MERGHH precipitation data from GPM |
| SRTM Void Filled | 90 m | - | Digital Elevation Model (DEM) | The 2D representation of topographic surface |

### 2.2.3 Model-based data

The Regional Data Assimilation and Prediction System (RDAPS; Davies et al., 2005) is one of the numerical weather forecast models used by the Korea Meteorological Administration, which is based on Unified Model (UM) by the United

5  Kingdom Met Office. The analysis-forecast products with about a hundred variables are generated with 12 km of spatial resolution and 70 vertical layers. They are provided four times a day (03:00, 09:00, 15:00, 21:00 KST) for 87-hour forecasts with 3-hour time steps. A total of 7 variables in UM RDAPS analysis data (i.e., temperature, dew-point temperature, RH, maximum wind speed, visibility at the height above the ground, and PBLH and surface pressure) were used as meteorological input variables in this study. These meteorological variables are commonly used to estimate ground-level PM

10  concentrations (Lv et al., 2017; He and Huang, 2018).

The Sparse Matrix Operator Kernel Emissions (SMOKE; Baek et al., 2009) is based on emission inventories generally provided as an annual total emission amount for each emission source. Hourly emission data with 9 km spatial resolution





were obtained from the National Institute of Environmental Research (NIER). Among the 47 chemical composition parameters in SMOKE outputs, 14 of PM-related emission data (i.e., ISOPRENE, TRP1, CH4, NO, NO2, NH3, HCOOH, HCHO, CO, SO2, POA, PNO3, PSO4 and PMFINE) were used in this study. The selected parameters are mostly those defined by Aerosol Emission 5 (AE5) which is one of the $PM_{2.5}$ chemical mechanisms (Jimenez et al, 2013) and major

precursors forming the PM (Xu et al., 2015b; van Zelm et al., 2016; Gao et al., 2016).

The Breathing Earth System Simulator (BESS; Ryu et al., 2018) is the MODIS-based model that couples atmosphere and canopy radiative transfers, photosynthesis, transpiration, and energy balance. It includes an atmospheric radiative transfer model and an ANN approach with MODIS atmospheric products. Daily BESS shortwave radiation products with 5 km spatial resolution were obtained from the Environmental Ecology Lab at Seoul National University

(http://environment.snu.ac.kr/bess_rad/).

### 2.2.4 Other input variables

Population density by region (obtained from the Statistical Geographic Information Service (SGIS; https://sgis.kostat.go.kr/)) and Day of Year (DOY) were used as additional input variables together with remote sensing and model-based meteorological and emission variables. Population density was calculated for each administrative division, in which a unit is

the number of people per square kilometre, and then converted to raster with a 1 km grid. In this study, DOY was converted to values ranging from -1 to 1 with a one-year period using a sine function considering seasonality (i.e., setting the middle of summer as 1 and the middle of winter as -1; Stolwijk et al., 1999). Road network data were not used in this study, as the use of the road data often yielded inaccurate results over non-urban areas in our preliminary analyses.

### 2.2.5 Data pre-processing

A total of 32 input variables from satellite and model-based data were used for the estimation of ground-level PM concentrations in the RF machine learning. All data collected at 13:00 KST were used to develop PM estimation models to match the acquisition time of MODIS Aqua aerosol products over the study area. The observed PM concentrations (i.e., target variables) were log-transformed because high concentration data were relatively small. To ensure the reliability of GOCI-derived aerosol products, the four rule-based filters used in Choi (2017) were applied: buddy check, local variance

check, sub-pixel cloud fraction check, and diurnal variation check. The same NDVI values during the interval of MODIS 16-days NDVI were used in the models. GPM precipitation data were converted into 24-hour accumulated precipitation data using 48 half-hourly data prior to the target time (i.e., hourly). UM RDAPS reanalysis data were linearly interpolated using analysis fields at 09:00 and 15:00 KST. DEM, urban area ratio and population density data were used as constant variables during the study period. Input data with different spatial resolutions were resampled to a 1 km MODIS grid using bilinear

interpolation. A total of 32 input variables and their abbreviations are summarized in Table 2.



**Table 2: List of input variables (and their abbreviations) used to estimate ground-level particulate matter concentrations.**

| Data | Variables | Abbreviations |
|---|---|---|
| Satellite-based remote sensing data | Aerosol Optical Depth | AOD |
| | Fine Mode Fraction | FMF |
| | Single Scattering Albedo | SSA |
| | Ångström Exponent | AE |
| | Normalized Difference Vegetation Index | NDVI |
| | Urban area ratio | Urban_ratio |
| | 24-hour Accumulated Precipitation | Precip |
| | Digital Elevation Model | DEM |
| Model-based meteorological data | Temperature at the height above ground | Temp |
| | Dew-point temperature at the height above ground | Dew |
| | Relative humidity at the height above ground | RH |
| | Pressure surface | P_srf |
| | 3-hour maximum wind speed at the height above ground | MaxWS |
| | Planetary Boundary Layer Height | PBLH |
| | Visibility at the height above ground | Visibility |
| | Solar Radiation | RSDN |
| Model-based emission data | ISOPRENE ($C_5H_8$) | ISOPRENE |
| | Monoterpene ($C_{10}H_{16}$) | TRP1 |
| | Methane ($CH_4$) | CH4 |
| | Nitric oxide (NO) | NO |
| | Nitrogen dioxide ($NO_2$) | NO2 |
| | Ammonia ($NH_3$) | NH3 |
| | Formic acid (HCOOH) | HCOOH |
| | Formaldehyde (HCHO) | HCHO |
| | Carbon monoxide (CO) | CO |
| | Sulfur dioxide ($SO_2$) | SO2 |
| | Primary organic aerosol | POA |
| | Primary nitrate | PNO3 |





| | Primary sulfate | PSO4 |
|---|---|---|
| | Other primary PM$_{2.5}$ | PMFINE |
| Ancillary data | Population density | PopDens |
| | Converted Day of Year | DOY |

## 3 Methodology

The process flow diagram for the estimation of ground-level PM concentrations is shown in Figure 2. The constructed data were divided into two groups by date: 80% of the data were used for model development and the remaining 20% were used for hindcast validation considering data distribution by PM concentration levels. The data for model development were again randomly divided into training (80%) and test (20%) datasets. Since PM reference data had a skewed distribution (i.e., a number of low concentration samples and a few high concentration samples), oversampling and subsampling approaches were conducted only for the training dataset to avoid over- or under-estimation due to biased sample distribution. Then, the RF machine learning method was applied to the training datasets to develop the models for estimating ground-level PM concentrations.

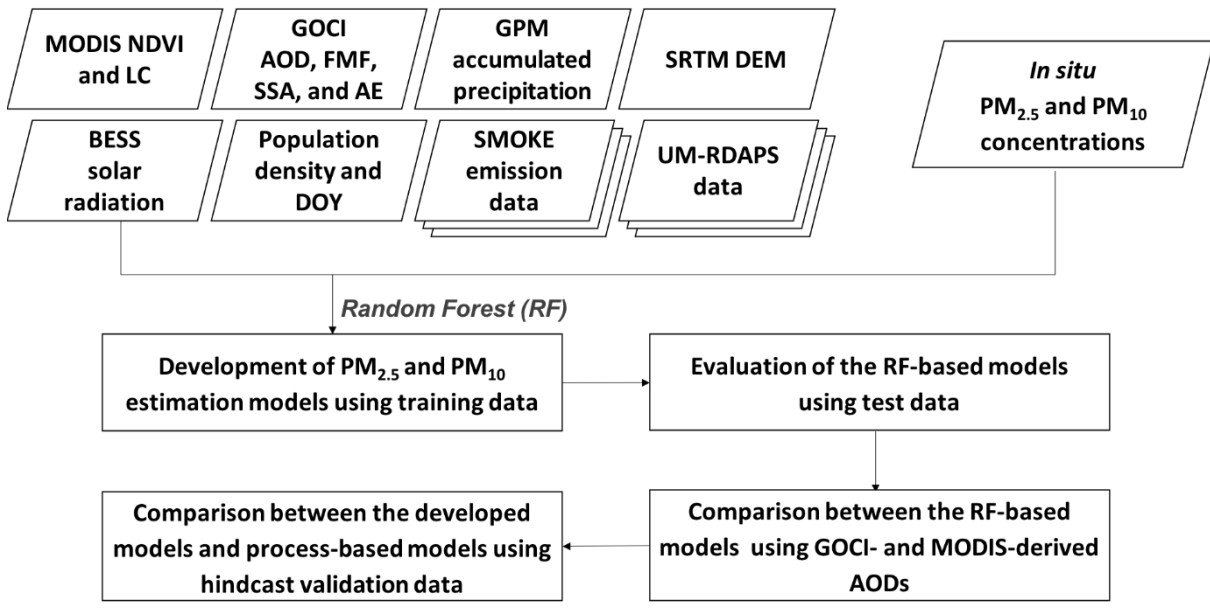

**Figure 2: Process flow diagram of the estimation of ground level particulate matter concentrations proposed in this study.**



## 3.1 Oversampling and Subsampling

Many of the in-situ observation data used in this study showed low concentrations, while there were a relatively small number of observations of high concentrations. This imbalance in samples could result in biased estimation with a significant underestimation of high concentration data. Thus, over- and sub-sampling approaches were conducted for the training

datasets to overcome the problem caused by the unbalanced samples (Table 3).

The oversampling approach is based on the assumption that the PM concentration of a training sample (i.e., at a pixel) is not significantly different from those of its neighbouring pixels. Input variables in the adjacent pixels of high concentration samples were extracted using 3 x 3 or 5 x 5 windows with the corresponding target variables (i.e., $PM_{2.5}$ and $PM_{10}$) randomly perturbed within 5% of the focus pixel concentrations. The subsampling approach was applied to the low concentration data

(e.g., 30-60 $\mu g/m^3$ for $PM_{10}$) that had too many samples compared to the other levels of concentrations.

**Table 3: The number of samples for training, test, and hindcast validation datasets. The adjusted sample size for training data was determined through the over-/sub-sampling approaches.**

|  | Training dataset | | Test dataset | Hindcast validation dataset |
|---|---|---|---|---|
|  | Original | Adjusted |  |  |
| $PM_{10}$ | 7919 | 14201 | 1545 | 3906 |
| $PM_{2.5}$ | 3038 | 5738 | 776 | 1364 |

## 3.2 Machine learning approach (Random Forest; RF)

Machine learning approaches have been widely used in various remote sensing studies with classification and regression (Liu et al., 2015; Ke et al., 2016; Lee et al., 2016; Hu et al., 2017b; Jang et al., 2017). Since RF has proved to be useful for remote sensing-based regression tasks (Jang et al.,2017; Chen et al., 2018; Yoo et al., 2018), it was used to develop models to estimate ground-level PM concentrations in this study. RF is an ensemble model based on classification and regression trees (CART) with randomized node optimization and bootstrap aggregating (aka bagging; Breiman, 2001). RF generates

numerous independent trees to overcome the limitations of a single decision (or regression) tree method, such as the dependency on a single tree and the problem of overfitting the training data. A multitude of independent trees are ensembled to reach a solution by majority voting for classification or averaging for regression. RF provides information on how a variable contributes to model development using out-of-bag (OOB) data that are not used in training a model (Breiman, 2001). When a variable from OOB data is randomly permuted, the change in mean square error in percentage is calculated.

The larger the increase in the error for a variable, the more contributing the variable is. RF was applied to the training data to develop the models for estimating ground-level PM concentrations. The models were evaluated using the test and hindcast validation data.





## 3.3 Model evaluation

Accuracy assessment of the developed models were conducted using the test and hindcast validation datasets based on the five metrics—coefficient of determination ($R^2$), RMSE, relative RMSE (rRMSE), mean bias (MB), and mean error (ME). rRMSE, MB, and ME are calculated as:

$$\text{rRMSE} = \frac{\text{RMSE}}{\bar{y}} \times 100 \text{ \%}, \tag{1}$$

$$\text{MB} = \frac{1}{N} \sum_{i=1}^{N} (f_i - y_i) \tag{2}$$

$$\text{ME} = \frac{1}{N} \sum_{i=1}^{N} |f_i - y_i| \tag{3}$$

where $y_i$ is the observed data, $\bar{y}$ is the mean of the observed data, $f_i$ is an estimated value, and $N$ is the number of observations. The rRMSE is the RMSE normalized by the mean value of observed data, which is useful for comparing results with different scales. The MB and ME are the average of variation between the model-derived and observed values, with the exception that ME uses only absolute difference. The MB presents a tendency of overestimation or underestimation by a model. The ME is the difference between observation and estimation (Boylan and Russell, 2006).

## 3.4 Comparison with other approaches

MODIS AOD is one of the widely used satellite-based aerosol products, which has often been used to estimate PM concentrations. The developed RF models were compared with those using MODIS AOD instead of GOCI aerosol products. Unlike GOCI, MODIS only provides AOD with 3 km resolution (i.e., MYD04_3K) over land, AOD was used for developing MODIS-based models without incorporating other aerosol-related variables (i.e., AE, FMF and SSA). In order to compare the performance between MODIS- and GOCI-based RF models, 50 % of the samples that were commonly included in both MODIS and GOCI datasets were used to develop the models, while the remaining data were used to validate the models.

In addition, the ground-level PM concentrations predicted using the GOCI-based RF models were compared to the simulated and predicted results by GEOS-Chem and CMAQ models. The comparison among the GOCI-based model, GEOS-Chem, and CMAQ to in situ measurements was conducted using the hindcast validation dataset. The results from the GOCI-based models were resampled to the GEOS-Chem grid with 0.25° x 0.3125° from January to September 2016 and CMAQ grids with 9 km x 9 km for 2015-2016 for comparison to in situ measurements, respectively. The approach by van Donkelaar et al. (2010) that uses the ratio between the ground-level data and total column of AOD to satellite-based AOD (i.e., here GOCI AOD) using the vertical profile of AOD from GEOS-Chem was adopted to predict ground-level PM concentrations (i.e., GOCI-GEOS-Chem fused PM estimation).



# 4 Results and discussion

## 4.1 Performance of the RF models

The evaluation results of the developed models for estimating $PM_{10}$ and $PM_{2.5}$ concentrations using the test datasets over South Korea are presented in Table 4. The models (the improved models hereafter) based on the balanced training samples

through over-/sub-sampling resulted in $R^2$ values of 0.78 and 0.73, and RMSEs of 17.08 µg/m³ and 8.25 µg/m³ for $PM_{10}$ and $PM_{2.5}$, respectively. There was a significant improvement in using the balanced training samples instead of the original samples (decrease of RMSE ~30% and rRMSE ~10%). MB and ME also confirmed that the balanced samples improved the models estimating ground level PM concentrations (Table 3; Figure 3). In particular, high concentration data (over 150 µg/m³ for $PM_{10}$ and 50 µg/m³ for $PM_{2.5}$) were well estimated by the improved models. The slopes of the trends were also

improved from 0.46-0.48 to 0.77-0.78. The slopes were still lower than 1, and it is due to the slight overestimation of low PM concentration data (Figure 3).

Although it is not possible to directly compare the present results with those from other studies, the results from this study agreed well with those from recent literature that used machine learning approaches for estimating PM concentrations (Gupta et al., 2009b; Wu et al., 2012; Li et al., 2017a; Yeganeh et al., 2017; Hu et al., 2017b; Chen et al., 2018). Hu et al. (2017b)

estimated surface $PM_{2.5}$ concentrations using RF, resulting in the cross validation $R^2$ of 0.8 and RMSE of 2.83 µg/m³. Similarly, Chen et al. (2018) compared three different methods (i.e., RF, generalized additive model (GAM), and non-linear exposure-lag-response model (NEM)) to estimate surface $PM_{2.5}$ concentrations over China during 2014-2016. Their results for daily estimation show cross validation $R^2$ of 0.83, 0.55, and 0.51 for RF, GAM, and NEM, respectively, implying the robustness of machine learning compared to traditional statistical models. A geographically adjusted deep belief network

(Geoi-DBN) was used to estimate $PM_{2.5}$ over China and showed a good correlation with observation data ($R^2 = 0.88$ and RMSE = 13.68 µg/m³; Li et al., 2017a). The literature shows that empirical models using statistical and machine learning approaches often underestimate high PM concentrations (Wu et al., 2012; Li et al., 2017a). However, the RF-based models developed in our study has proved to be effective for modelling high ground-level PM concentrations.

**Table 4: Accuracy assessment results of the RF-based models for estimating PM concentrations using the test datasets during**
**2015-2016.**

| | $R^2$ | RMSE [a] (µg/m³) | rRMSE [b] (%) | MB [c] (µg/m³) | ME [d] (µg/m³) | Slope | Intercept |
|---|---|---|---|---|---|---|---|
| Model (with original training samples) | | | | | | | |
| $PM_{10}$ | 0.58 | 24.34 | 36.96 | -5.24 | 15.41 | 0.48 | 28.94 |
| $PM_{2.5}$ | 0.59 | 10.53 | 36.46 | -2.30 | 7.37 | 0.46 | 13.30 |
| Improved model (with balanced training samples) | | | | | | | |
| $PM_{10}$ | 0.78 | 17.08 | 25.94 | 2.93 | 12.78 | 0.78 | 17.16 |



| | | | | | | | |
|---|---|---|---|---|---|---|---|
| PM$_{2.5}$ | 0.73 | 8.25 | 28.58 | 1.71 | 6.18 | 0.77 | 8.30 |

[a] Root Mean Square Error; [b] Relative Root Mean Square Error; [c] Mean Bias; [d] Mean Error

In addition, the seasonal variation of model performance for 2015 and 2016 is shown in Table 5. The $R^2$ values for PM$_{10}$ estimations are the highest (0.87) in winter with an RMSE of 12.78 µg/m³ and the lowest (0.50) in summer with an RMSE of

5    12.62 µg/m³, as compared to $R^2$ values of 0.77 and 0.74 with RMSEs of 16.61 µg/m³ and 13.07 µg/m³ in fall and spring, respectively. The summer season resulted in relatively high rRMSE for estimating ground-level PM concentrations compared to the other seasons. This is mainly because ground-level PM concentrations are typically low in summer in South Korea. The relatively small sample size in summer and cloud contamination might lead to estimation errors (Shi et al., 2014; Sogacheva et al., 2017).

**Table 5: Seasonal variation of model performance for estimating particulate matter (PM) concentrations. Spring, summer, fall, and winter correspond to March to May, June to August, September to November, and December to February, respectively.**

| | | $R^2$ | RMSE [a] (µg/m³) | rRMSE [b] (%) | MB [c] (µg/m³) | ME [d] (µg/m³) | Slope | Intercept |
|---|---|---|---|---|---|---|---|---|
| PM$_{10}$ | Annual | 0.76 | 13.04 | 19.32 | 3.09 | 9.83 | 0.75 | 19.78 |
| | Spring | 0.74 | 13.07 | 17.77 | 3.08 | 9.98 | 0.70 | 25.06 |
| | Summer | 0.50 | 12.62 | 28.88 | 0.33 | 9.23 | 0.48 | 22.95 |
| | Fall | 0.77 | 16.61 | 26.69 | 7.76 | 11.81 | 0.87 | 15.76 |
| | Winter | 0.87 | 12.78 | 19.22 | 3.71 | 9.20 | 0.87 | 12.29 |
| PM$_{2.5}$ | Annual | 0.82 | 5.92 | 18.90 | 1.36 | 4.42 | 0.81 | 7.21 |
| | Spring | 0.82 | 5.90 | 19.01 | 1.14 | 4.47 | 0.75 | 8.77 |
| | Summer | 0.63 | 7.79 | 30.98 | 3.15 | 6.20 | 0.61 | 12.97 |
| | Fall | 0.85 | 8.12 | 27.50 | 3.89 | 6.53 | 0.88 | 7.30 |
| | Winter | 0.79 | 7.94 | 20.99 | 0.72 | 5.56 | 0.82 | 7.65 |

[a] Root Mean Square Error; [b] Relative Root Mean Square Error; [c] Mean Bias; [d] Mean Error;







**Figure 3: The model test results of daily PM$_{10}$ and PM$_{2.5}$ estimation: (a) PM$_{10}$ estimation model using the original samples, (b) PM$_{10}$ estimation using the balanced samples through over-/ sub-sampling, (c) PM$_{2.5}$ estimation model using the original samples, and (d) PM$_{2.5}$ estimation model using the balanced samples through over-/ sub-sampling.**

5    Figure 4 depicts the top 10 input variables that were identified as the most contributing variables by the improved RF models

for estimating PM$_{10}$ and PM$_{2.5}$ concentrations. The results indicate that AOD, DOY, MaxWS, RSDN, and Dew (i.e., dew-

point temperature) were commonly identified as contributing variables by the RF models to estimate both ground-level PM$_{10}$





and PM$_{2.5}$ concentrations. The AOD was identified as the most significant factor, which agreed well with the exiting literature (Yu et al., 2017; Zang et al., 2017; Chen et al., 2018). Although most high PM concentration samples had high AOD values, some high PM samples had low AOD values. Careful examination of the samples shows that there were Asian dust events at high altitudes during the period of study, which did not affect ground-level PM concentrations. This could be

an error source, implying that altitude information of such dust events can be used to further improve the models for estimating ground-level PM concentrations.

Some meteorological variables indicating the atmospheric conditions also contributed to the estimation of ground-level PM concentrations in the improved models. There is a relationship between solar radiation and aerosols in which solar radiation increases with decreasing aerosol concentration (Préndez et al., 1995; Hu et al., 2017a; Borlina and Rennó, 2017). Prior

studies noted that there is an inverse relationship between wind speed and both PM$_{10}$ and PM$_{2.5}$ (Gupta et al., 2006; Maraziotis et al., 2008; Krynicka and Drzeniecka-Osiadacz, 2013). This relationship causes an increase in PM concentrations under low wind speed conditions but a decrease under high wind speed conditions, which is also confirmed in the present study. This means that atmospheric conditions such as congestion have significant impacts on surface PM concentrations. The results correspond to previous studies (e.g., You et al., 2015; Yeganeh et al., 2017; Hu et al., 2017b; Yu

et al., 2017) showing that meteorological factors are strongly effective in improving PM estimation models. Interestingly, the anthropogenic factors such as LC_ratio (urban ratio), PopDens (population density), NH3, and SO2 were more important for PM$_{2.5}$ estimation than PM$_{10}$. This implies that the sources of PM$_{2.5}$ are mainly anthropogenic in South Korea (Moon et al., 2011; gon Ryou et al., 2018).





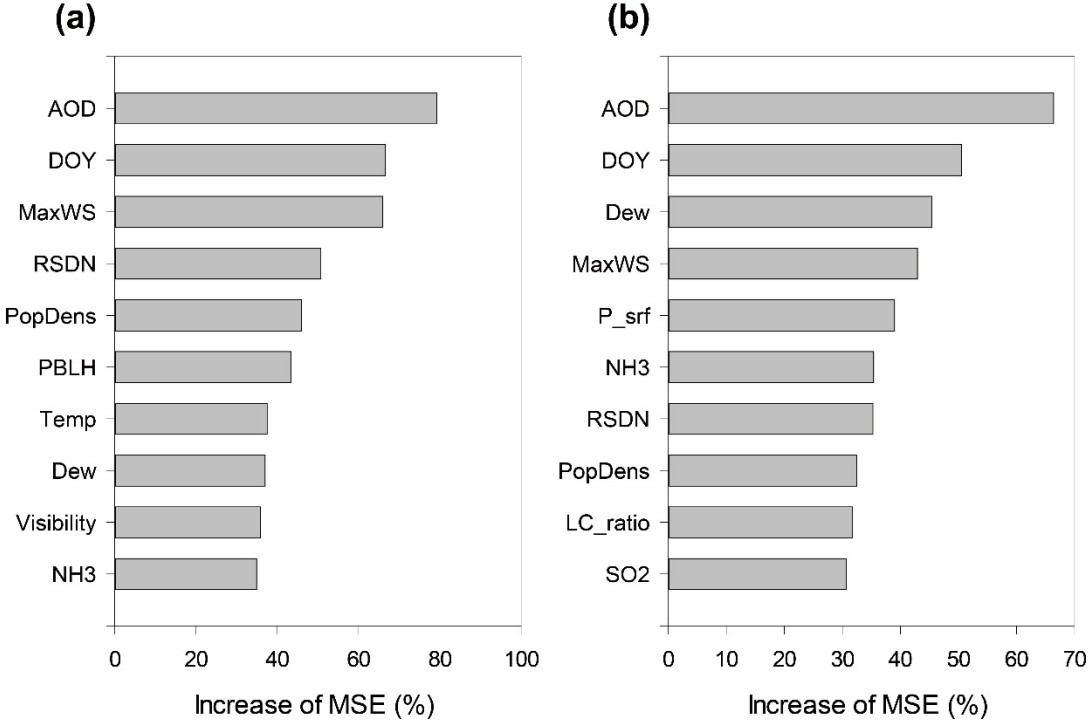

**Figure 4: Variable importance of the top 10 input variables identified by the random forest models for estimating ground-level (a) PM$_{10}$ and (b) PM$_{2.5}$ concentrations.**

## 4.2 Spatial distribution of PM concentrations using the improved RF models

Figure 5 illustrates the spatial distribution of 2-year (2015-2016) averaged surface PM$_{10}$ and PM$_{2.5}$ concentrations at 1 km resolution with station-based in-situ PM$_{10}$ and PM$_{2.5}$ concentrations over South Korea. The pixels that have concentration values for more than 5 % of the period (> 36 days for the two years) were used to produce the spatial distribution maps to secure the reliability of the distribution. Thus, the maps have some no data pixels. The predicted PM$_{10}$ and PM$_{2.5}$ have similar spatial patterns with relatively high concentrations for urban areas especially around metropolitan areas, and agree well with observed concentrations (Figure 5).

The seasonal maps of PM$_{10}$ and PM$_{2.5}$ concentrations are also shown in Figure 6. South Korea has the rainy season usually in June and July. For this reason, cloud contaminants are much more significant in the summer season than the other seasons, which resulted in many no data pixels for the summer maps (Figure 6). The ground-level PM concentrations in the spring and winter are much higher than in summer and fall for PM$_{10}$. The results agree well with the general seasonal patterns of



PM10 concentrations of South Korea, where PM concentrations are much higher in spring due to Asian dust inflow carried by westerly winds (Park et al., 2017). In addition, anthropogenic emissions generally increase PM concentrations in winter (Lu et al., 2011; Li et al., 2016). The seasonal distribution of PM2.5 concentrations is similar to that of PM10. However, high concentrations were predominantly found in fall for PM2.5. The cold Siberian high pressure might explain this. When warm

5 air from the south flows into the study area, and while the force of the Siberian anticyclone stops, an inversion layer is formed. Then, PM is trapped because the atmospheric circulation becomes stagnant. Another reason can be explained by the relative overestimation of PM2.5 by the RF model in the fall season (Table 5). MB was greatest for the fall season among the four seasons indicating overestimation of PM2.5. A more careful data configuration between training and test samples with larger sample size may mitigate such an overestimation.





**Figure 5: Maps of two-year averaged particulate matter concentrations: (a) PM$_{10}$ by the RF model, (b) *in situ* PM$_{10}$ at stations, (c) PM$_{2.5}$ by the RF model, and (d) *in situ* PM$_{2.5}$ at stations.**

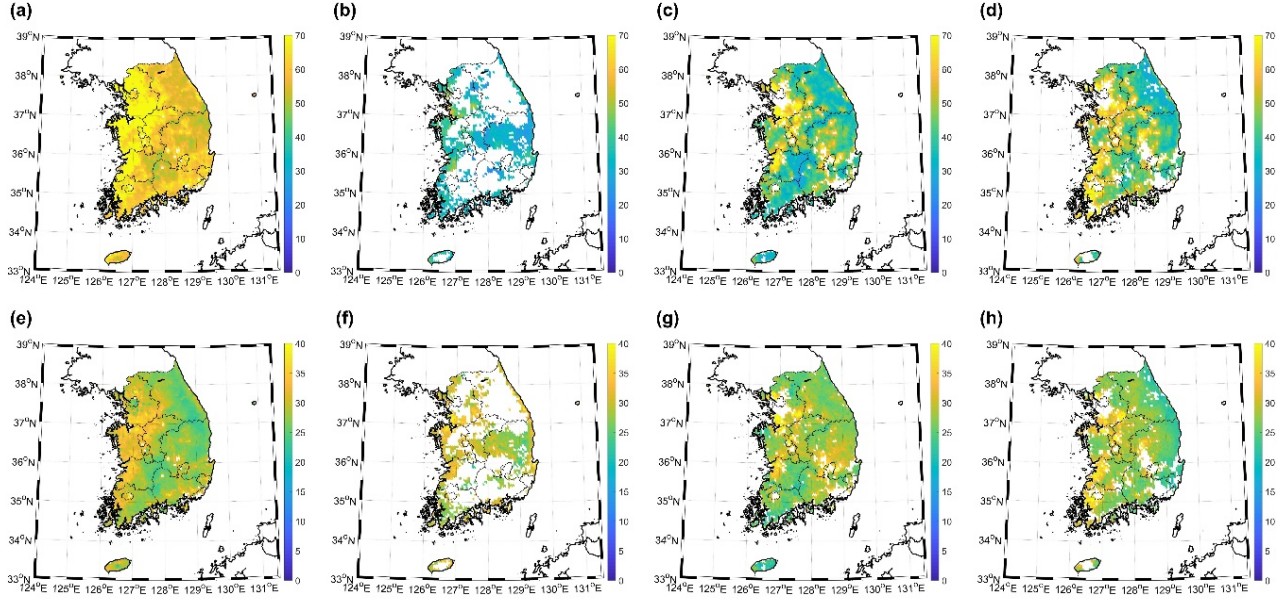

**Figure 6: Spatial distributions of seasonal mean particulate matter concentrations: (a) PM$_{10}$ for spring, (b) PM$_{10}$ for summer, (c)**
**PM$_{10}$ for fall, and (d) PM$_{10}$ for winter, (e) PM$_{2.5}$ for spring, (f) PM$_{2.5}$ for summer, (g) PM$_{2.5}$ for fall, and (h) PM$_{2.5}$ for winter.**

## 4.3 Comparison of ground PM concentrations based on GOCI and MODIS AODs

The existing studies have generally used MODIS-derived AOD to estimate surface PM concentrations for various countries because of its global coverage and high quality (Remer et al., 2006; Gupta et al., 2009a,b; Van Donkelaar et al., 2010; Wang et al., 2010; Chudnovsky et al., 2014; You et al., 2015; Hu et al., 2017b; Yu et al., 2017; He and Huang, 2018). In this section, the estimated ground-level PM$_{10}$ and PM$_{2.5}$ concentrations are compared based on GOCI AOD and MODIS AOD. Figure 7 displays the scatterplots showing the cross-validation results of the RF-based models using GOCI-derived and MODIS-derived AODs. Although there was no statistically significant difference between the two types of models through ANOVA tests, the GOCI-based RF models produced slightly better accuracy metrics (i.e., R$^2$, RMSE, and rRMSE) than MODIS -based RF models for estimating ground-level PM concentrations. Considering the advantages of GOCI as a geostationary satellite sensor (i.e., high spatial and temporal resolutions; 8 times a day with a 500 m grid size), it is very promising to use GOCI-derived products as input to PM estimation models. It should also be noted that GOCI-2, which has enhanced sensor specifications (i.e., 10 data collection per day at 250 m spatial resolution) is planned to be launched in 2019.





**Figure 7: Scatterplots between the estimated and observed particulate matter concentrations: (a) by the MODIS-based RF model for PM$_{10}$, (b) by the GOCI-based RF model for PM$_{10}$, (c) by the MODIS-based RF model for PM$_{2.5}$, and (d) by the GOCI-based RF model for PM$_{2.5}$.**




## 4.4 Comparison with the process-based models

The RF-based models for estimating ground-level PM$_{10}$ and PM$_{2.5}$ concentrations were further compared with process-based models, i.e., GEOS-Chem and CMAQ. Some studies investigated the GEOS-Chem simulated PM and AOD by integrating satellite-derived AOD to improve their results (Van Donkelaar et al., 2010; Van Donkelaar et al., 2015; Xu et al., 2015a).

Figure 8 shows the comparison of the accuracy metrics of the three models: the GEOS-Chem simulated, GOCI-GEOS-Chem fused, and the RF-predicted PM concentrations using the hindcast validation datasets (Table 3). The GOCI-GEOS-Chem fused PM$_{10}$ concentration have less errors than the GEOS-Chem simulated PM$_{10}$ concentration, which agrees well with the existing literature. However, both tend to significantly underestimate the ground-level PM$_{10}$ concentration when compared to the proposed RF model. Although the GOCI-GEOS-Chem fused PM$_{2.5}$ concentration shows higher RMSE and mean error

than GEOS-Chem PM$_{2.5}$ concentration due to overestimation, the R$^2$ and slope of the GOCI-GEOS-Chem fused PM$_{2.5}$ concentration improved when compared to those of the GEOS-Chem PM$_{2.5}$ concentration. The RF models also produced better performance than CMAQ for estimating both PM$_{10}$ and PM$_{2.5}$ concentrations (Figure 9). Similar to the GEOS-Chem models, CMAQ tends to underestimate PM concentrations showing a large negative MB value.

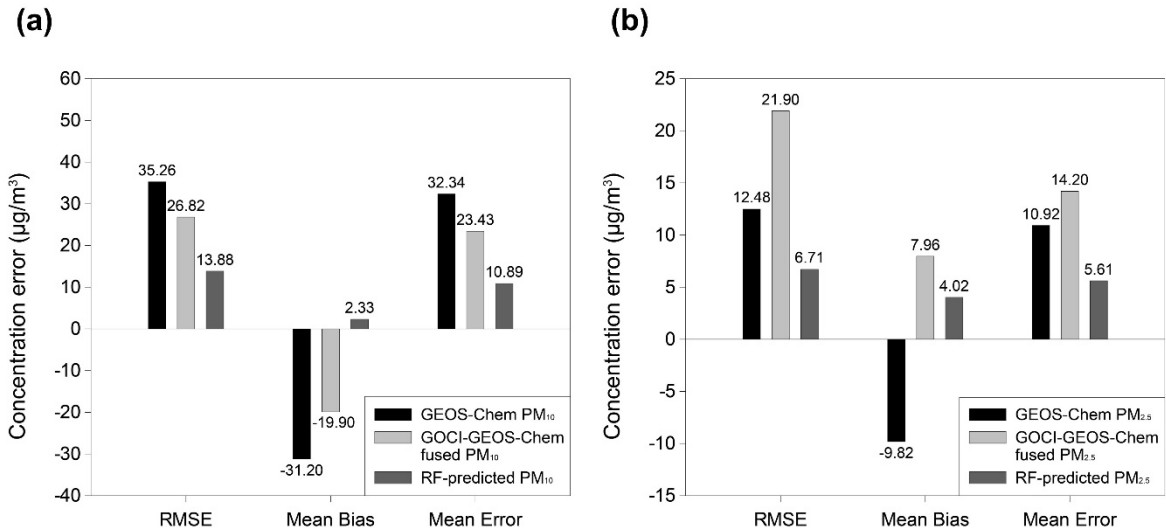

**Figure 8: Comparison of the three models (i.e., GEOS-Chem based, GOCI-GEOS-Chem fused, and the present RF-based models) using the hindcast validation data for estimating particulate matter concentrations: (a) PM$_{10}$ and (b) PM$_{2.5}$ with Root Mean Square Error (RMSE), Mean Bias (MB), and Mean Error (ME).**





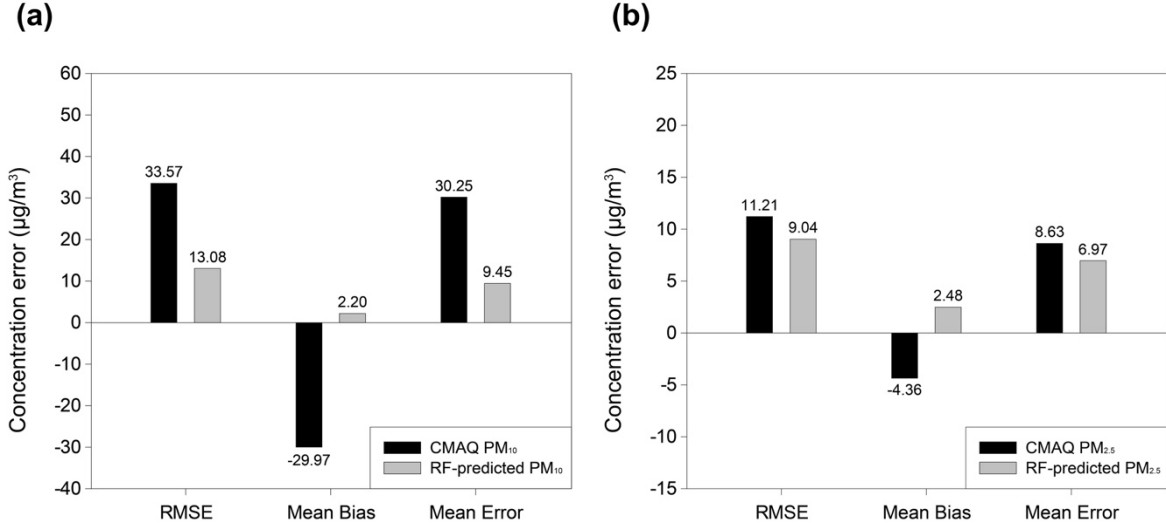

**Figure 9: Comparison between the RF-based and CMAQ models using the hindcast validation data for estimating particulate matter concentrations: (a) PM$_{10}$ and (b) PM$_{2.5}$ with Root Mean Square Error (RMSE), Mean Bias (MB), and Mean Error (ME).**

## 5 Conclusions

In this study, machine learning (i.e., RF) based models were developed to estimate ground-level PM$_{10}$ and PM$_{2.5}$ concentrations through the synergistic use of satellite data and model output over South Korea. The RF-based models developed using the balanced training samples produced good performance resulting in R$^2$ values of 0.78 and 0.73, and RMSEs of 17.08 µg/m$^3$ and 8.25 µg/m$^3$ for PM$_{10}$ and PM$_{2.5}$, respectively. In particular, the proposed models estimated high PM concentrations well. GOCI-derived AOD was identified as the most significant input variable for estimating ground-

level PM concentrations. A few meteorological variables such as MaxWS, RSDN, and dew-point temperature were also revealed as contributing variables. In addition, the anthropogenic factors such as urban ratio, population density, emission of SO$_2$ and NH$_3$ were considered significant for estimating PM$_{2.5}$ concentrations. Two-year and seasonal averaged maps of ground level PM concentrations agree with spatio-temporal patterns of PM concentrations reported in the literature.

The proposed RF models were also compared to the two process-based models (GEOS-Chem and CMAQ) using the

hindcast validation data. When GOCI-derived AOD was incorporated with the GEOS-Chem data, the estimation of PM concentrations improved. However, the incorporated approach still underestimated high concentrations, when compared to the proposed RF models. Similar results were found for the comparison between the RF models and CMAQ, which implies the robustness of the proposed approach.

Although the proposed models performed better than the existing models, there are several ways to further improve the

proposed models, which deserve further investigation. First, more input variables, especially those that are related to vertical



information of AOD, can be used to improve the models. In addition, other sophisticated approaches such as deep learning could be utilized to improve the estimation accuracy for ground-level PM concentrations. Although only two-year data were used in this study, longer archives can be used to further refine the models. The synergistic use of forthcoming geostationary satellite series of GEO-KOMPSAT (GK)-2A with Advanced Meteorological Imager (AMI) and GK-2B with GOCI-II and

Geostationary Environment Monitoring Spectrometer (GEMS) sensors, will provide more accurate aerosol information with higher spatial and temporal resolutions than those of GOCI. Such a synergy is likely to improve the estimation of ground-level PM concentrations in the near future.

**Acknowledgments**

This study was supported by a grant from the National Institute of Environmental Research (NIER), funded by the Ministry of Environment (MOE) of the Republic of Korea (NIER-2017-01-02-063), the Space Technology Development Program through the National Research Foundation of Korea (NRF) funded by the Ministry of Science, ICT, & Future Planning (NRF-2017M1A3A3A02015981), and the National Strategic Project-Fine Particle of the National Research Foundation of Korea (NRF) funded by the Ministry of Science and ICT (MSIT), the Ministry of Environment (ME), and the Ministry of

Health and Welfare (MOHW) (NRF-2017M3D8A1092021).

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
