# Peer review of "Estimation of ground-level particulate matter concentrations through the synergistic use of satellite observations and process-based models over South Korea"

_Atmospheric Chemistry and Physics, 2018_

## Referee Comment (RC1) · Anonymous Referee #1 · 24 Sep 2018

General comments

This paper develops a statistical model to estimate ground-level PM over Korea from AOD retrieved by the GOCI geostationary satellite instrument. It is interesting to see how well a statistical model can estimate PM from geostationary AOD. This study also presents a thorough comparison to other commonly used approaches like AOD from MODIS vs GOCI and simulations by CTMs vs statistical models. The paper is generally clear, although it can be more concise and better organized. I recommend it for publication in ACP after the following questions are well addressed.

Major comments:

1.  In Fig 8, the GOCI-GEOS-Chem fused $PM_{2.5}$ seems to have considerably larger bias than even just the GEOS-Chem simulated $PM_{2.5}$, which is not what I was expecting since GOCI AOD provides an observational constraint to the simulation. Could you please explain why GOCI-GEOS-Chem fused $PM_{2.5}$ has such a large bias? Also you mentioned in text P20L10 that $R^2$ and slope of the GOCI-GEOS-Chem fused $PM_{2.5}$ is better than the GEOS-Chem $PM_{2.5}$. Could you please put them into numbers?

2.  The description of data used in this study needs to be more detailed. For example, for observations of PM, suggest to include general description of how PM is measured, like what instrument is used.  For GEOS-Chem and CMAQ simulations, you should include model version, what meteorological fields are used, what emissions are used to help readers interpret results.

3.  I didn't understand why the RF PM in Fig 8 was different from that in Fig 9. Could you please explain what makes the difference?

4.  I'm surprised to see that the PM estimates from MODIS are basically the same as they are from GOCI given that MODIS provides about 8 times less data than GOCI and that MODIS cannot resolve the diurnal variation of AOD yet GOCI can. I was expecting that MODIS will have at least larger variability than GOCI but it is not the case either in Fig 7. Could you please explain why GOCI is not showing a pronounced enhancement over MODIS here?

5.  As you've shown in Fig 4 that meteorological parameters have playing an important role in relating AOD to PM2.5 as well. Could you please comment on the accuracy of these parameters?

6.  You mentioned in the paper that RF well estimates PM high concentrations. Could you please elaborate what findings support this statement? And why RF is good in estimating high concentrations but not small? What potential bias in RF can be reflected from this finding?

7.  P11L19. Could you please make clear if the remaining aerosol variables (AE, FMF, etc) come from GOCI? If so, they are given at different temporal resolutions from MODIS AOD. Could you please justify this?

Minor comments:

1.  Suggest to distinguish subplots (e.g., Fig 3, 4, 5, 6, 7, 8, 9) by adding titles and remove letter like (a), (b), so readers don't need to refer to the caption to find what each subplots means. Then captions can be more concise.
2.  In Fig 3, what do different colors in circles mean?
3.  P2L10-17: I didn't understand how the example of Zang et al., (2017) supported the statement of the limitation of ground-based measurements.
4.  P2L31: I suggest to move CTM to the next paragraph to have this paragraph focused on statistical method. Also this is a good place to introduce RF. Maybe moving the introduction of RF starting from P10L15 to this paragraph.
5.  P5 section 2.2.2 and 2.2.3: suggest to describe what data a satellite or model is used to provide to this study as the very first sentence when introducing a new model. It can be very confusing given so many different models and names all introduced in this section.
6.  P7L23, please put "high concentration" into numbers.
7.  P11L16, "MODIS only provides AOD with 3 km resolution". Could you please verify whether MODIS AOD is 3 km? I think it should be 10 km.
8.  P13L8, why summer sample size is small? If it's due to cloud contamination, then should be swap the order of cloud contamination and small sample size in the sentence.
9.  P15L3-4: It seems that the example of Asia dust events at high altitudes is used to support the case of high PM and low AOD. I think it's actually conflicting to the statement. I'd expect high altitude dusts contribute to high AOD yet low surface PM. Could you please explain?
10. P15L8: "in which solar radiation increases with decreasing aerosol concentration", do you mean solar radiation reaching the surface increases with …?
11. P18L16, I got an impression that the author seemed to be overemphasizing the 500m resolution in the paper. However, all GOCI data used in this study are aggregated to 6km, so suggest to change 500 m resolution to 6 km and change high spatial resolution to moderate spatial resolution. Also please change the spatial resolution about GOCI aerosol products to 6km elsewhere too.

Technical comments:

1.  P1L1: should be "ground-level"
2.  P1L16: "The long exposure" should be "Long-term exposure".
3.  P1L29: "the proposed RF MODELS yielded better performance" than what?
4.  P2L3: "especially PM10 and PM2.5". This is where the abbreviation PM should be introduced, not at line 11.
5.  P2L17: cut "on the other hand".
6.  P5L7: should be "at" 400nm and 870nm.
7.  P5L8: change "MODIS" to "MODIS satellite instrument".
8.  P5L9: cut "to observe the Earth's environment".
9.  P5L10, change "and Aerosol…" to ", Aerosol"
10. P7L4, change "which is one of the … and" to "as"
11. P16L9, cut " Thus, .. data pixels".

12. P16L13, cut "season".
13. P20L3-4, suggest to cut "Some studies investigated … Xu et al., 2015a) to avoid repetition to the introduction.

---

## Referee Comment (RC2) · Anonymous Referee #2 · 10 Oct 2018

The authors developed a random forest model to predict ground-level PM10 and PM2.5 based on geostationary satellite observations and model-based emission and meteorological outputs over South Korea. Oversampling and subsampling strategies were used to balance the training samples in order to better estimate high-level PM concentrations. This study contributes to more accurate PM predictions by appropriately adjusting the reference PM data and the synergistic use of satellite AOD data and other environmental variables. The manuscript is well written and is suitable for publication in Atmospheric Chemistry and Physics after the following comments are addressed.

Comments:

1. Line 29, Page 2: The sentence starts with "more recent studies …" needs references about these studies.
2. Line 7, Page 3: can you add references to support the statement that many studies have focused on PM prediction in the United States because of less cloud cover in satellite data?
3. Section 3.1: it is not clear how the authors chose the sizes of the sampling window (e.g., $3 \times 3$ and $5 \times 5$)? Was there any sensitivity analysis being conducted to determine appropriate window sizes? In addition, it was likely that an increased adjusted sample size (after oversampling and subsampling) contributed to better modeling performance. So, more clarifications are needed to better explain the effectiveness of the oversampling and subsampling strategies.
4. Section 3.2: Did the authors test the correlation among the independent variables? If two or more independent variables are correlated with each other, the model may include more variables than it is necessary, which could lead to bias/uncertainty in the interpretation of the results.
5. Table 5: I suggest adding a column showing the sample sizes (N) of the models.
6. Line 13, Page 15: For the word "congestion", did the authors mean "advection" or "convection"?

---

## Referee Comment (RC3) · Anonymous Referee #3 · 7 Nov 2018

This study proposed improved random forest model to predict ground surface PM concentrations over South Korea. Multiple satellite-derived products and model-based meteorological parameters were used as input variables. The results showed that the improved model is effective in predicting high PM concentrations compared to previous research. The manuscript is overall complete, well written and ready for publishing after the following comments are addressed.

1. section 2.2, please explain why you chose those variables as explanatory indicators.

[Figure]

2. The authors adopted oversampling and under-sampling strategies to alleviate the biased estimation problem. "Input variables in the adjacent pixels of high concentration samples were extracted using 3 x 3 or 5 x 5 windows with the corresponding target variables (i.e., PM2.5 and PM10) randomly perturbed within 5% of the focus pixel concentrations. " Will this perturbation introduce uncertainty? How do you chose appropriate window size?

3. In page 12 line 24, "However, the RF-based models developed in our study has proved to be effective for modelling high ground-level PM concentrations." Could you explain why the RF-based models in this study is more effective than previous studies? Is that because sampling strategies used in your study? If so, could you compare the model performances with and without your sampling strategies?

4. Could you explain the accuracy of MODIS-derived AOD and GOCI-derived AOD? This may help explain why GOCI-AOD-based models outperformed MODIS-AOD-based models.

---

## Author Comment (AC1) · 2 Dec 2018

**Authors' responses (ACP-2018-647)**

The authors would like to thank the editors and the reviewers for their precious time and invaluable comments. The corresponding changes and refinements are highlighted in yellow in the revised paper and are also summarized in our responses below. Authors' responses are in blue. Reviewer's comments are in black. When the manuscript is cited, it is shown in italics.

**Reviewer # 1**

**Major comments:**

1. In Fig 8, the GOCI-GEOS-Chem fused $PM_{2.5}$ seems to have considerably larger bias than even just the GEOS-Chem simulated $PM_{2.5}$, which is not what I was expecting since GOCI AOD provides an observational constraint to the simulation. Could you please explain why GOCI-GEOS-Chem fused $PM_{2.5}$ has such a large bias? Also you mentioned in text P20L10 that $R^2$ and slope of the GOCI-GEOS-Chem fused $PM_{2.5}$ is better than the GEOS-Chem $PM_{2.5}$. Could you please put them into numbers?

   → The slope and $R^2$ values were additionally added in the text for clarity. In fact, the bias of GOCI-GEOS-Chem fused $PM_{2.5}$ was less than the bias of GOCI-Chem simulated $PM_{2.5}$. However, the RMSE and $R^2$ of GOCI-GEOS-Chem fused $PM_{2.5}$ were larger than those of GEOS-Chem simulated $PM_{2.5}$ due to overestimation.

   → P21L12: *"Consequently, the proposed RF models have the lowest RMSE, MB, and ME among those models. Although the results of GOCI-GEOS-Chem fused $PM_{2.5}$ showed that $R^2$ (GEOS-Chem $PM_{2.5}$: 0.00, GOCI-GEOS-Chem fused $PM_{2.5}$: 0.14) and slope (GEOS-Chem $PM_{2.5}$: -0.02, GOCI-GEOS-Chem fused $PM_{2.5}$: 1.41) improved more than those of GEOS-Chem $PM_{2.5}$, the RMSE, MB, and ME of the fused model were higher than the GEOS-Chem model because the fused model overestimated PM concentrations."*

2. The description of data used in this study needs to be more detailed. For example, for observations of PM, suggest to include general description of how PM is measured, like what instrument is used. For GEOS-Chem and CMAQ simulations, you should include model version, what meteorological fields are used, what emissions are used to help readers interpret results.

   → Additional descriptions for observations of PM, GEOS-Chem, and CMAQ were added.

   → P5L10: *"PMs at stations are measured based on a beta attenuation monitoring (BAM) technique which is widely used for automatic air monitoring (Zhan et al., 2017; Zhao et al., 2016). The measurement results are expressed as mass concentration per unit volume (i.e., µg/m³) converted to room temperature (20 °C, 1 atm)."*

   → P12L26: *"The GEOS-Chem v10-01 was utilized with the Global Forecast System (GFS; produced by the National Centres for Environmental Prediction (NCEP)) as meteorological fields, and MIX Asian emission inventory as emissions. The CMAQ model version 4.7.1 was used to simulate the ground-level $PM_{10}$ and $PM_{2.5}$ concentrations. Meteorological fields simulated by the Weather Research and Forecasting (WRF) model and emission data from the SMOKE model were utilized to run the CMAQ model."*

**Authors' responses (ACP-2018-647)**

3.  I didn't understand why the RF PM in Fig 8 was different from that in Fig 9. Could you please explain what makes the difference?

    ➔ The RF-predicted PM was resampled to match the spatial resolution with each process-based model (i.e., GEOS-Chem (0.25° x 0.3125°), CMAQ (9 km X 9 km)). The considered dates were also different in order to compare the results on the same scale (GEOS-Chem: from January to September 2016, CMAQ: 2015-2016).

4.  I'm surprised to see that the PM estimates from MODIS are basically the same as they are from GOCI given that MODIS provides about 8 times less data than GOCI and that MODIS cannot resolve the diurnal variation of AOD yet GOCI can. I was expecting that MODIS will have at least larger variability than GOCI but it is not the case either in Fig 7. Could you please explain why GOCI is not showing a pronounced enhancement over MODIS here?

    ➔ GOCI provides data eight times a day, but the model developed in this paper only considered the GOCI aerosol data at 04:00 UTC, when MODIS data are available.

    ➔ The benefit of using GOCI lies in its high temporal resolution (8 times a day). Choi et al. (2018) showed that the accuracy of GOCI AOD is comparable (slightly higher than) to that of MODIS AOD.

    ➔ Supplementary Figure 2 shows the comparison between observed PM concentrations and satellite-derived AODs. The left column depicts the relationship between PM concentrations and MODIS-derived AODs. The right column displays the relationship between PM concentrations and GOCI-derived AODs. There is slightly higher correlation between PMs and GOCI-derived AODs than between PMs and MODIS-derived AODs.

    ➔ The GOCI(V2)-derived AOD was compared with MODIS-derived DT/DB AOD using AERONET AOD over the GOCI coverage region (Table 2 in Choi et al., 2018). These comparisons showed similar results in terms of R, MB, and RMSE. We only focused on the land AOD comparison in the red box because we used the AOD over the land area.

**Authors' responses (ACP-2018-647)**

[Figure]

*Supplementary Figure 2: Comparison of PM concentrations to MODIS-derived AOD (left column) and GOCI-derived AOD (right column), (a) comparison between PM₁₀ and MODIS-derived AOD, (b) comparison between PM₁₀ and GOCI-derived AOD, (c) comparison between PM₂.₅ and MODIS-derived AOD, (d) comparison between PM₂.₅ and GOCI-derived AOD.*

**Table 2.** Statistics of land and ocean AOD comparisons between AERONET/SONET and satellite products, as shown in Fig. 3.

| Satellite AOD algorithm | $N$ | $R$ | MB | $f$ within $EE_{DT}$ | RMSE |
|---|---|---|---|---|---|
| *Land AOD comparison with AERONET* | | | | | |
| GOCI YAER V1 all QA | 47 850 | 0.86 | −0.015 | 0.49 | 0.24 |
| GOCI YAER V1 QA3 | 38 183 | 0.92 | −0.066 | 0.49 | 0.18 |
| GOCI YAER V2 | 45 643 | 0.91 | 0.010 | 0.60 | 0.16 |
| MODIS DT | 3228 | 0.92 | 0.043 | 0.62 | 0.18 |
| MODIS DB | 3463 | 0.93 | 0.007 | 0.73 | 0.16 |
| *Land AOD comparison with SONET* | | | | | |
| GOCI YAER V1 all QA | 12 974 | 0.83 | −0.048 | 0.45 | 0.29 |
| GOCI YAER V1 QA3 | 10 483 | 0.88 | −0.103 | 0.42 | 0.27 |
| GOCI YAER V2 | 12 238 | 0.86 | −0.021 | 0.51 | 0.24 |
| MODIS DT | 733 | 0.82 | 0.104 | 0.46 | 0.29 |
| MODIS DB | 1258 | 0.86 | 0.000 | 0.67 | 0.27 |
| *Ocean AOD comparison with AERONET* | | | | | |
| GOCI YAER V1 all QA | 19 945 | 0.83 | 0.056 | 0.55 | 0.17 |
| GOCI YAER V1 QA3 | 18 308 | 0.88 | 0.043 | 0.62 | 0.13 |
| GOCI YAER V2 | 18 499 | 0.89 | 0.008 | 0.71 | 0.11 |
| MODIS DT | 680 | 0.92 | 0.033 | 0.73 | 0.09 |

**Authors' responses (ACP-2018-647)**

5.  As you've shown in Fig 4 that meteorological parameters have playing an important role in relating AOD to $PM_{2.5}$ as well. Could you please comment on the accuracy of these parameters?

    → Figure 4 shows the variable importance of the top 10 input variables identified by the random forest models for estimating ground-level $PM_{10}$ and $PM_{2.5}$ concentrations. Random forest ranks the relative importance of each variable using out-of-bag (OOB) data which are not used in training the model. The OOB error is defined as the difference between the actual value of data and the prediction result from each tree. The importance of the i-th variable is defined as the average of difference between OOB error from original dataset and replaced dataset for all trees. It should be noted that the relative importance is local, and thus the results do not mean that the top identified variables globally contributed to the model.

    → Unfortunately, the official accuracy of the parameters themselves from the numerical model was not available. The key meteorological variables contributing to the estimation of the ground level PM concentrations were discussed in the text, including the wind speed, radiation, and visibility.

    → P16L8-19: *"Some meteorological variables indicating the atmospheric conditions also contributed to the estimation of ground-level PM concentrations in the improved models. There is a relationship between solar radiation and aerosols in which solar radiation reaching the surface increases with decreasing aerosol concentration (Préndez et al., 1995; Hu et al., 2017a; Borlina and Rennó, 2017). Prior studies noted that there is an inverse relationship between wind speed and both $PM_{10}$ and $PM_{2.5}$ (Gupta et al., 2006; Maraziotis et al., 2008; Krynicka and Drzeniecka-Osiadacz, 2013). This relationship causes an increase in PM concentrations under low wind speed conditions but a decrease under high wind speed conditions, which is also confirmed in the present study. This means that atmospheric conditions such as air stagnation have significant impacts on surface PM concentrations. The results correspond to previous studies (e.g., You et al., 2015; Yeganeh et al., 2017; Hu et al., 2017b; Yu et al., 2017) showing that meteorological factors are strongly effective in improving PM estimation models. Interestingly, the anthropogenic factors such as LC_ratio (urban ratio), PopDens (population density), $NH_3$, and $SO_2$ were more important for $PM_{2.5}$ estimation than $PM_{10}$. This implies that the sources of $PM_{2.5}$ are mainly anthropogenic in South Korea (Moon et al., 2011; gon Ryou et al., 2018)."*

6.  You mentioned in the paper that RF well estimates PM high concentrations. Could you please elaborate what findings support this statement? And why RF is good in estimating high concentrations but not small? What potential bias in RF can be reflected from this finding?

    → When using the balanced training samples (i.e., through over-/sub-sampling), the accuracy of the RF models significantly improved, especially for high concentration samples. The comparison of the model performances with and without the sampling strategies is shown in Table 4.   Good performance for high concentrations does not

mean poor performance for low concentrations. Of course, since the RF models were trained using more balanced samples (similar sample size of low and high concentrations), they produced much better performance than those with the original training samples (biased sample size: much large for low concentrations). However, as you can see, there was slight accuracy decrease for the test samples of low concentrations (refer to the following table). However, such a small decrease can be compromised with dramatic accuracy increase for high concentrations.

➔ We added an explanation of this potential bias toward low concentration samples.
➔ P13L17: *"This significant improvement in the estimation performance was mainly due to the proposed sampling strategies in order to use more balanced training data. The use of the balanced training data resulted in the huge increase of the estimation accuracy of ground-level PM concentrations especially for high concentration samples at the compensation of slight accuracy decrease for low concentrations."*

| PM10_original | | | | | | | |
|---|---|---|---|---|---|---|---|
| Interval | R2 | RMSE | rRMSE | MB | ME | slope | intercept |
| All | 0.58 | 24.34 | 36.96 | -5.24 | 15.41 | 0.48 | 28.94 |
| <=30 (low) | 0.10 | 15.15 | 64.54 | 11.62 | 12.00 | 0.58 | 21.48 |
| >150 (high) | 0.00 | 102.19 | 52.16 | -88.53 | 88.53 | 0.01 | 104.74 |
| PM10_oversamples | | | | | | | |
| Interval | R2 | RMSE | rRMSE | MB | ME | slope | intercept |
| All | 0.78 | 17.08 | 25.94 | 2.93 | 12.78 | 0.78 | 17.16 |
| <=30 (low) | 0.08 | 18.46 | 78.64 | 13.95 | 14.23 | 0.64 | 22.35 |
| >150 (high) | 0.74 | 29.27 | 14.94 | -17.69 | 18.56 | 0.88 | 6.61 |
| PM2.5_original | | | | | | | |
| Interval | R2 | RMSE | rRMSE | MB | ME | slope | intercept |
| All | 0.59 | 10.53 | 36.46 | -2.30 | 7.37 | 0.46 | 13.30 |
| <=15 (low) | 0.03 | 8.21 | 70.10 | 6.68 | 6.77 | 0.26 | 15.30 |
| >75 (high) | 0.02 | 47.91 | 51.05 | -44.40 | 44.40 | 0.11 | 38.74 |
| PM2.5_oversamples | | | | | | | |
| Interval | R2 | RMSE | rRMSE | MB | ME | slope | intercept |
| All | 0.73 | 8.25 | 28.58 | 1.71 | 6.18 | 0.77 | 8.30 |
| <=15 (low) | 0.03 | 9.90 | 84.55 | 7.93 | 8.03 | 0.33 | 15.73 |
| >75 (high) | 0.96 | 5.29 | 5.64 | -3.88 | 4.29 | 1.13 | -15.77 |

7. P11L19. Could you please make clear if the remaining aerosol variables (AE, FMF, etc) come from GOCI? If so, they are given at different temporal resolutions from MODIS

AOD. Could you please justify this?

➔ The word *"remaining data"* was replaced with *"remaining samples"* to avoid confusion. *"Remaining samples"* indicate the samples excluding the training samples from each dataset for MODIS- and GOCI-based RF models, not aerosol variables that come from GOCI. As mentioned in the text, GOCI-based RF model is developed focused on 13:00 KST when the acquisition time of MODIS Aqua aerosol products over the study area (See P7L21-Section 2.2.5).

➔ P12L24: *"… while the remaining samples were used to validate the models."*

**Authors' responses (ACP-2018-647)**

**Minor comments:**

1. Suggest to distinguish subplots (e.g., Fig 3, 4, 5, 6, 7, 8, 9) by adding titles and remove letter like (a), (b), so readers don't need to refer to the caption to find what each subplots means. Then captions can be more concise.

   → Revised as suggested. Titles for subplots were added in figures and captions were more concise than before.

2. In Fig 3, what do different colors in circles mean?

   → The color scheme in Figure 3 is the point density of the scatterplots. We added this to the figure caption.

   → *"The colour scheme from blue to red indicates the point density: The blue point means low density while the red point shows high density."*

3. P2L10-17: I didn't understand how the example of Zang et al., (2017) supported the statement of the limitation of ground-based measurements.

   → The Zang et al. (2017) briefly talked about the limitation of using station parameters in their paper. We additionally mentioned the following sentence.

   → P2L16: *"Their study suggested an inversion model to estimate $PM_{2.5}$ but showed a limitation in that the model can only be used in areas near ground stations, which are required by the model to derive its parameters."*

4. P2L31: I suggest to move CTM to the next paragraph to have this paragraph focused on statistical method. Also this is a good place to introduce RF. Maybe moving the introduction of RF starting from P10L15 to this paragraph.

   → Revised as suggested. We also added more description for clarity.

   → P2L29-P3L10: *"Chemical transport models (CTM) have also been combined with satellite observations to estimate ground-level PM concentrations. To estimate global 6-year (2001-2006) averaged $PM_{2.5}$ concentrations, Van Donkelaar et al. (2010) combined Moderate Resolution Imaging Spectroradiometer (MODIS) and MISR-derived AODs, and multiplied them by the ratio between $PM_{2.5}$ and AOD simulated by the GEOS-Chem model (i.e., CTM). Their results showed a strong spatial agreement with in-situ $PM_{2.5}$ concentrations in North America (slope = 1.07; $R^2$ = 0.59).*
   *More recent studies explored advanced statistical and machine learning approaches to improve the prediction of ground-level PM concentrations by deploying mixed-effects models, geographically weighted regression (GWR), support vector machines (SVM), or artificial neural networks (ANN) (Gupta et al., 2009b; You et al., 2015; Li et al., 2017a; Chen et al., 2018). Machine learning approaches have been widely used in various remote sensing studies thanks to their flexibility with classification and regression (Im et al., 2009; Lu et al., 2011a, Liu et al., 2015; Ke et al., 2016; Pham et al., 2017; Forkuor et al., 2018). In particular, random forest (RF) has proved to be useful for remote sensing-based regression tasks (Yoo et al., 2012;*

*Jang et al., 2017; Richardson et al., 2017; Yoo et al., 2018). To estimate daily PM$_{2.5}$ concentrations over the United States, Hu et al. (2017b) incorporated MODIS AOD, simulated GEOS-Chem AOD, meteorological data, and land-use information in an RF model."*

5. P5 section 2.2.2 and 2.2.3: suggest to describe what data a satellite or model is used to provide to this study as the very first sentence when introducing a new model. It can be very confusing given so many different models and names all introduced in this section.

   ➔ The summary of the satellite and model data used was added at the very beginning of each section.
   ➔ P5L15: *"Various remote sensing data were used in this study such as GOCI aerosol products, MODIS Normalized Difference Vegetation Index (NDVI), land cover product, Global Precipitation Measurement (GPM) 30-min precipitation data, and the Shuttle Radar Topography Mission (SRTM) elevation data."*
   ➔ P7L3: *"Along with satellite-based data, the outputs from three models were combined. The three models were: the Regional Data Assimilation and Prediction System (RDAPS), the Sparse Matrix Operator Kernel Emissions (SMOKE), and the Breathing Earth System Simulator (BESS)."*

6. P7L23, please put "high concentration" into numbers.

   ➔ Actually, we considered the log-transformation because the observed PM concentration has large concentration range. The large concentration range leads to underestimate at high concentration due to lack of high concentration samples in the dataset. For this reason, we changed the sentence to the following.
   ➔ P8L9: *"The observed PM concentrations (i.e., target variables) were log-transformed because the concentration range is large and has a positively skewed distribution."*

7. P11L16, "MODIS only provides AOD with 3 km resolution". Could you please verify whether MODIS AOD is 3 km? I think it should be 10 km.

   ➔ MODIS provides the AOD product with both 10 km and 3 km resolutions.

8. P13L8, why summer sample size is small? If it's due to cloud contamination, then should be swap the order of cloud contamination and small sample size in the sentence.

   ➔ It is because of cloud contamination, so that sentence is changed following your comment.
   ➔ P14L10: *"The cloud contamination and the relatively small sample size in summer, might lead to estimation errors."*

9. P15L3-4: It seems that the example of Asia dust events at high altitudes is used to support the case of high PM and low AOD. I think it's actually conflicting to the statement. I'd expect high altitude dusts contribute to high AOD yet low surface PM.

Could you please explain?

➔ There was some mistake in explaining the result and we modified the sentence.
➔ P16L3: *"Careful examination of the samples shows that there were Asian dust events at low altitudes in those cases, which were not effectively included in the AOD derived from satellite sensor systems. In other words, the satellite-derived AOD has a weak sensitivity in capturing aerosols at low altitudes (Choi et al. 2018)."*

10. P15L8: "in which solar radiation increases with decreasing aerosol concentration", do you mean solar radiation reaching the surface increases with …?

➔ Yes. We revised the text accordingly.
➔ P16L10: *"… in which solar radiation reaching the surface increases with decreasing aerosol concentration."*

11. P18L16, I got an impression that the author seemed to be overemphasizing the 500m resolution in the paper. However, all GOCI data used in this study are aggregated to 6km, so suggest to change 500 m resolution to 6 km and change high spatial resolution to moderate spatial resolution. Also please change the spatial resolution about GOCI aerosol products to 6km elsewhere too.

➔ GOCI aerosol products used in this study have 6 km resolution although GOCI L1B data are provided 8 times a day with 500 m of spatial resolution. We accepted your suggestion and revised as following.
➔ P19L16-P20L2: *"Considering the advantages of GOCI as a geostationary satellite sensor (i.e., moderate spatial and temporal resolutions; 8 times a day with a 6 km grid size of the aerosol product), it is very promising to use GOCI-derived products as input to PM estimation models. It should also be noted that GOCI-2, which has enhanced sensor specifications (i.e., 10 data collections per day at 3 km spatial resolution of the aerosol product) is planned to be launched in 2019."*

**Authors' responses (ACP-2018-647)**

**Technical comments:**

1. P1L1: should be "ground-level"

   → Revised as suggested.
   P1L1: *"Estimation of ground-level particulate matter concentrations…"*

2. P1L16: "The long exposure" should be "Long-term exposure".

   → Revised as suggested.
   P1L16: *"Long-term exposure to particulate matter (PM) with…"*

3. P1L29: "the proposed RF MODELS yielded better performance" than what?

   → The proposed RF models showed better performance than the process-based approaches (i.e., GEOS-Chem and CMAQ). Revised accordingly.

4. P2L3: "especially $PM_{10}$ and $PM_{2.5}$". This is where the abbreviation PM should be introduced, not at line 11.

   → The abbreviation PM is relocated at P2L3 for describing $PM_{10}$ and $PM_{2.5}$ instead at line 11.
   → P2L3: *"… especially $PM_{10}$ and $PM_{2.5}$ (particulate matter (PM) with an aerodynamic diameter…"*
   → P2L11: *"… in providing spatially continuous PM concentrations that…"*

5. P2L17: cut "on the other hand".

   → We removed it as suggested.

6. P5L7: should be "at" 400nm and 870nm.

   → Revised as suggested.
   P5L22: *"… and Ångström exponent (AE) at 440 and 870 nm with…"*

7. P5L8: change "MODIS" to "MODIS satellite instrument".

   → Revised as suggested.
   P5L24: *"MODIS satellite instrument, onboard the Terra and Aqua satellites…"*

8. P5L9: cut "to observe the Earth's environment".

   → We removed it as suggested.

**Authors' responses (ACP-2018-647)**

9. P5L10, change "and Aerosol…" to ", Aerosol"

   ➔ Revised as suggested.
   P5L25: *"with 1 km resolution (MYD13A2; Solano et al., 2010), Aerosol 5-min L2 swath data…"*

10. P7L4, change "which is one of the … and" to "as"

    ➔ Revised as suggested.
    P7L17: *"The selected parameters are mostly those defined by Aerosol Emission 5 (AE5) as major precursors forming the PM…"*

11. P16L9, cut " Thus, .. data pixels".

    ➔ Revised as suggested.

12. P16L13, cut "season".

    ➔ Revised as suggested.

13. P20L3-4, suggest to cut "Some studies investigated … Xu et al., 2015a) to avoid repetition to the introduction.

    ➔ Revised as suggested.

[revised manuscript text omitted]

---

## Author Comment (AC2) · 2 Dec 2018

The authors would like to thank the editors and the reviewers for their precious time and invaluable comments. The corresponding changes and refinements are highlighted in yellow in the revised paper and are also summarized in our responses below. Authors' responses are in blue. Reviewer's comments are in black. When the manuscript is cited, it is shown in italics.

**Reviewer # 2**

**Comments:**

1. Line 29, Page 2: The sentence starts with "more recent studies …" needs references about these studies.

   ➔ We added the references.
   ➔ P3L3: *"More recent studies explored advanced statistical and machine learning approaches to improve the prediction of ground-level PM concentrations by deploying mixed-effects models, geographically weighted regression (GWR), support vector machines (SVM), or artificial neural networks (ANN) (Gupta et al., 2009b; You et al., 2015; Li et al., 2017a; Chen et al., 2018)."*

2. Line 7, Page 3: can you add references to support the statement that many studies have focused on PM prediction in the United States because of less cloud cover in satellite data?

   ➔ Since we could not find the references that exactly discussed the cloud cover issue in different regions, we removed the sentence and rephrase the paragraph.
   ➔ P3L12: *"Most previous studies have mainly used AOD produced from polar orbiting satellite sensor systems such as MODIS and MISR. They provide AOD worldwide but only make it available once a day because of the revisit time. A major problem with daily AOD is cloud contamination. Therefore, it is difficult to obtain spatially continuous AOD over cloudy regions such as East Asia in summer monsoon."*

3. Section 3.1: it is not clear how the authors chose the sizes of the sampling window (e.g., 3×3 and 5×5)? Was there any sensitivity analysis being conducted to determine appropriate window sizes? In addition, it was likely that an increased adjusted sample size (after oversampling and subsampling) contributed to better modeling performance. So, more clarifications are needed to better explain the effectiveness of the oversampling and subsampling strategies.

   ➔ The window sizes for oversampling were determined considering the distribution of the samples through the examination of the histograms of the PM concentrations. We tested various combinations of sampling size for over-/sub- sampling, and then determined the appropriate sizes.
   ➔ The pixels within the window were ordered based on the proximity to the center (refer to Supplementary Figure 1). For example, oversampling for pixels of an interval might be conducted for first three pixels following the order, while oversampling for pixels of another interval might be conducted for up to the 13th pixel within the window.

**Authors' responses (ACP-2018-647)**

➔ We explained the process in detail as suggested.

➔ P10L9: *"The pixels within a circular window with a radius of 3 pixels (i.e., 37 pixels including the focus cell) were considered as potential neighbouring pixels with sorting the proximity to the centre (see Supplementary Figure 1 ). First, the intervals of 30 µg/m³ and 20 µg/m³ were applied to the $PM_{10}$ and $PM_{2.5}$ samples, respectively. The second groups (i.e., 30-60 µg/m³ for $PM_{10}$ and 20-40 µg/m³ for $PM_{2.5}$) had the largest sample size, and thus the subsampling approach based on simple random sampling (i.e., 50%) was applied to the second groups. For the other groups, we multiplied an integer value ranging from 1 to 37 by the sample size of each group to produce a more balanced sample distribution (i.e., the smaller the sample size, the larger the integer), and then oversampling based on the ordered neighbouring pixels was performed. Input variables in the adjacent pixels of high concentration samples were extracted with the corresponding target variables (i.e., $PM_{2.5}$ and $PM_{10}$) randomly perturbed within 5% of the focus pixel concentrations. This oversampling approach can effectively reduce underestimation of high PM concentrations resulting from the small training sample size of high concentration data."*

*Supplementary Figure 1: The pixels within the circular neighbouring window with a radius of 3 pixels considered for oversampling. The number in each pixel indicates the order of inclusion of the pixel for oversampling. For example, oversampling for pixels of an interval might be conducted for first three pixels following the order, while oversampling for pixels of another interval might be conducted for up to the 13th pixel within the window.*

➔ Figure R1 shows the histograms of original and adjusted samples.

[Figure]

**Figure R1: Histogram of original and adjusted samples for (a) PM₁₀ and (b) PM₂.₅**

4.  Section 3.2: Did the authors test the correlation among the independent variables? If two or more independent variables are correlated with each other, the model may include more variables than it is necessary, which could lead to bias/uncertainty in the interpretation of the results.

    ➔ We did test the correlation among the independent variables when we used statistical linear regression models. However, the multicollinearity has no effect on random forest models as random forest does not require any assumptions for variables. This is because each node of each tree is constructed by the values of the parameters sampled independently. Thus, only one response predictor is examined at once. Of course, random forest through variable selection considering the multicollinearity sometimes results in better performance. However, we did not see any significant difference in performance when using different input variables through feature selection.

5.  Table 5: I suggest adding a column showing the sample sizes (N) of the models.

**Authors' responses (ACP-2018-647)**

➔ Revised as suggested.

| | | $R^2$ | RMSE [a] ($\mu g/m^3$) | rRMSE [b] (%) | MB [c] ($\mu g/m^3$) | ME [d] ($\mu g/m^3$) | Slope | Intercept | Sample sizes (N) |
|---|---|---|---|---|---|---|---|---|---|
| $PM_{10}$ | Annual | 0.76 | 13.04 | 19.32 | 3.09 | 9.83 | 0.75 | 19.78 | 18466 |
| | Spring | 0.74 | 13.07 | 17.77 | 3.08 | 9.98 | 0.70 | 25.06 | 13132 |
| | Summer | 0.50 | 12.62 | 28.88 | 0.33 | 9.23 | 0.48 | 22.95 | 928 |
| | Fall | 0.77 | 16.61 | 26.69 | 7.76 | 11.81 | 0.87 | 15.76 | 1564 |
| | Winter | 0.87 | 12.78 | 19.22 | 3.71 | 9.20 | 0.87 | 12.29 | 2842 |
| $PM_{2.5}$ | Annual | 0.82 | 5.92 | 18.90 | 1.36 | 4.42 | 0.81 | 7.21 | 7188 |
| | Spring | 0.82 | 5.90 | 19.01 | 1.14 | 4.47 | 0.75 | 8.77 | 4510 |
| | Summer | 0.63 | 7.79 | 30.98 | 3.15 | 6.20 | 0.61 | 12.97 | 712 |
| | Fall | 0.85 | 8.12 | 27.50 | 3.89 | 6.53 | 0.88 | 7.30 | 961 |
| | Winter | 0.79 | 7.94 | 20.99 | 0.72 | 5.56 | 0.82 | 7.65 | 1005 |

6. Line 13, Page 15: For the word "congestion", did the authors mean "advection" or "convection"?

➔ We replaced "congestion" with "air stagnation" to improve clarity.

[revised manuscript text omitted]

---

## Author Comment (AC3) · 2 Dec 2018

**Authors' responses (ACP-2018-647)**

The authors would like to thank the editors and the reviewers for their precious time and invaluable comments. The corresponding changes and refinements are highlighted in yellow in the revised paper and are also summarized in our responses below. Authors' responses are in blue. Reviewer's comments are in black. When the manuscript is cited, it is shown in italics.

**Reviewer # 3**

**Comments:**

1. Section 2.2, please explain why you chose those variables as explanatory indicators.

   ➔ We chose the variables based on the recent literature. Many previous studies used these variables as predictors for estimating PM concentrations. For example, PM concentration is highly related to the AOD which provides a measure of the amount of aerosols in an atmospheric column. It is also affected by meteorological conditions and emissions (Van Donkelaar et al., 2015). NDVI and land cover information were found as effective predictors for air pollutant concentration in previous studies (Chudnovsky et al., 2014; Yeganeh et al, 2017). We added two sentences in Section 2.2 and more detailed description in the following subsections.

   ➔ P5L2: *"Data used in this study are ground observations as the target variable, and remote sensing data, model-based data, and other ancillary spatial data as explanatory variables. We selected the explanatory variables considering the recent literature that estimated ground PM concentrations (He and Huang, 2018; Chen et al., 2018; Brokamp et al., 2018), which are explained in the following sections."*

   ➔ *"…PMs at stations are measured based on a beta attenuation monitoring (BAM) technique which is widely used for automatic air monitoring (Zhan et al., 2017; Zhao et al., 2016). The measurement results are expressed as mass concentration per unit volume (i.e., $\mu g/m^3$) converted to room temperature (20 °C, 1 atm)…. Various remote sensing data were used in this study such as GOCI aerosol products, MODIS Normalized Difference Vegetation Index (NDVI), land cover product, Global Precipitation Measurement (GPM) 30-min precipitation data, and the Shuttle Radar Topography Mission (SRTM) elevation data. …. Along with satellite-based data, the outputs from three models were combined. The three models were: the Regional Data Assimilation and Prediction System (RDAPS), the Sparse Matrix Operator Kernel Emissions (SMOKE), and the Breathing Earth System Simulator (BESS). …"*

2. The authors adopted oversampling and under-sampling strategies to alleviate the biased estimation problem. "Input variables in the adjacent pixels of high concentration samples were extracted using 3 x 3 or 5 x 5 windows with the corresponding target variables (i.e., PM$_{2.5}$ and PM$_{10}$) randomly perturbed within 5% of the focus pixel concentrations. " Will this perturbation introduce uncertainty? How do you chose appropriate window size?

   ➔ We assumed that the value of neighboring pixels of each station is similar with the ground measurement PM concentrations within the 5% margin of error. We tested several perturbation percentages and found that there was no significant difference in

the results when low rates (up to 7%) were used. In other words, the estimation accuracy of high concentrations quite improved, while that of low concentration did not decrease much. Of course, uncertainty could introduced by conducting oversampling. However, such uncertainty can be negligible when using low perturbation rates.

➔ The window sizes for oversampling were determined considering the distribution of the samples through the examination of the histograms of the PM concentrations. We tested various combinations of sampling size for over-/sub- sampling, and then determined the appropriate sizes.

➔ The pixels within the window were ordered based on the proximity to the center (refer to Supplementary Figure 1). For example, oversampling for pixels of an interval might be conducted for first three pixels following the order, while oversampling for pixels of another interval might be conducted for up to the 13th pixel within the window.

➔ We added more detailed explanation about this process in the revision.

➔ P10L9: *"The pixels within a circular window with a radius of 3 pixels (i.e., 37 pixels including the focus cell) were considered as potential neighbouring pixels with sorting the proximity to the centre (see Supplementary Figure 1). First, the intervals of 30 $\mu g/m^3$ and 20 $\mu g/m^3$ were applied to the $PM_{10}$ and $PM_{2.5}$ samples, respectively. The second groups (i.e., 30-60 $\mu g/m^3$ for $PM_{10}$ and 20-40 $\mu g/m^3$ for $PM_{2.5}$) had the largest sample size, and thus the subsampling approach based on simple random sampling (i.e., 50%) was applied to the second groups. For the other groups, we multiplied an integer value ranging from 1 to 37 by the sample size of each group to produce a more balanced sample distribution (i.e., the smaller the sample size, the larger the integer), and then oversampling based on the ordered neighbouring pixels was performed. Input variables in the adjacent pixels of high concentration samples were extracted with the corresponding target variables (i.e., $PM_{2.5}$ and $PM_{10}$) randomly perturbed within 5% of the focus pixel concentrations. This oversampling approach can effectively reduce underestimation of high PM concentrations resulting from the small training sample size of high concentration data."*

*Supplementary Figure 1: The pixels within the circular neighbouring window with a radius of 3 pixels considered for oversampling. The number in each pixel indicates the order of inclusion of the pixel for*

**Authors' responses (ACP-2018-647)**

*oversampling. For example, oversampling for pixels of an interval might be conducted for first three pixels following the order, while oversampling for pixels of another interval might be conducted for up to the 13th pixel within the window.*

3. In page 12 line 24, "However, the RF-based models developed in our study has proved to be effective for modelling high ground-level PM concentrations." Could you explain why the RF-based models in this study is more effective than previous studies? Is that because sampling strategies used in your study? If so, could you compare the model performances with and without your sampling strategies?

   → The sampling strategies adopted in this study are one of the reasons that the RF-based models produced better than the existing models (Table 4; Figures 8 and 9). We also evaluated several other machine learning approaches such as support vector regression and artificial neural networks, but they did not produce better performance than the RF-based models. The other models that we compared in this study were physical model-based ones. The flexibility of the machine learning models might be another reason of their better performance than the existing ones.

   → The comparison of the model performances with and without the sampling strategies is shown in Table 4.

4. Could you explain the accuracy of MODIS-derived AOD and GOCI-derived AOD? This may help explain why GOCI-AOD-based models outperformed MODIS-AOD-based models.

   → Supplementary Figure 2 shows the comparison between observed PM concentrations and satellite-derived AODs. The left column depicts the relationship between PM concentrations and MODIS-derived AODs. The right column displays the relationship between PM concentrations and GOCI-derived AODs. There is slightly higher correlation between PMs and GOCI-derived AODs than between PMs and MODIS-derived AODs.

   → The GOCI(V2)-derived AOD was compared with MODIS-derived DT/DB AOD using AERONET AOD over the GOCI coverage region (Table 2 in Choi et al., 2018). These comparisons showed similar results in terms of R, MB, and RMSE. We only focused on the land AOD comparison in the red box because we used the AOD over the land area.

**Authors' responses (ACP-2018-647)**

[Figure]

*Supplementary Figure 2: Comparison of PM concentrations to MODIS-derived AOD (left column) and GOCI-derived AOD (right column), (a) comparison between PM₁₀ and MODIS-derived AOD, (b) comparison between PM₁₀ and GOCI-derived AOD, (c) comparison between PM₂.₅ and MODIS-derived AOD, (d) comparison between PM₂.₅ and GOCI-derived AOD.*

**Authors' responses (ACP-2018-647)**

**Table 2.** Statistics of land and ocean AOD comparisons between AERONET/SONET and satellite products, as shown in Fig. 3.

[revised manuscript text omitted]

---

## Author Response (AR2)

The authors would like to thank the editor for his precious time and invaluable comments. The corresponding changes and refinements are highlighted in yellow in the revised paper and are also summarized in our responses below. Authors' responses are in blue. Editor's comments are in black. When the manuscript is cited, it is shown in italics.

**Minor comments:**

1. What is the domain used for the models? Any boundary conditions used? Please add short info on both meteo and GEOSCHEM models.

   → Thanks for the comment. We added the information about the domain and boundary conditions of RDAPS and GEOS-Chem.

   → P6L7-9: *"The domain of the RDAPS is 77.38 ºE – 176.56 ºE and 9.59ºN – 61.27ºN. The RDAPS takes the information of initial and boundary conditions from UM - Global Data Assimilation and Prediction System (GDAPS) with the spatial resolution of 25 km x 25 km."*

   → P13L11-13: *"The nested domain for the GEOS-Chem simulation is 70ºE - 150ºE and 15ºN - 55ºN, which covers East Asia. The horizontal resolution of the nested model is 0.25º x 0.3125º. The boundary conditions for the nested model are from the GEOS-Chem global simulation at 2º x 2.5º horizontal resolution."*

2. The explanation of the balanced model is not easy to understand. page 10, lines 8-19. Can you try to reformulate. "intervals are applied"? "sorting the proximity"? In table 3 you refer to an adjusted dataset. What is that really?

   → We revised the paragraph to improve the readability.

   → P10L13-P11L8: *"The oversampling approach is based on the assumption that the PM concentration of a training sample (i.e., at a pixel) is not significantly different from those of its neighbouring pixels. The pixels within a circular window with a radius of 3 pixels (i.e., 37 pixels including the focus cell) were considered as potential neighbouring pixels (see Supplementary Figure 1). Those 37 neighbouring pixels were numbered based on the proximity to the centre (i.e., the closer the pixel is to the centre, the lower the number considering the direction from the focus). In order to perform oversampling, the intervals of 30 $\mu g/m^3$ and 20 $\mu g/m^3$ were first applied to the $PM_{10}$ and $PM_{2.5}$ samples, respectively (i.e., 0-30 $\mu g/m^3$, 30-60 $\mu g/m^3$,…, 360-390 $\mu g/m^3$, and >390 $\mu g/m^3$ for $PM_{10}$, and 0-20 $\mu g/m^3$, 20-40 $\mu g/m^3$, …, 100-120 $\mu g/m^3$, > 120 for $PM_{2.5}$). The second groups (i.e., 30-60 $\mu g/m^3$ for $PM_{10}$ and 20-40 $\mu g/m^3$ for $PM_{2.5}$) had the largest sample sizes, and thus the subsampling approach based on simple random sampling (i.e., 50%) was applied to the second groups. For the other groups, we multiplied an integer value ranging from 1 to 37 by the sample size of each group to produce a more balanced sample distribution (i.e., the smaller the sample size, the larger the integer). Oversampling was then performed based on the order of the neighbouring pixels*

*was performed. Input variables in the adjacent pixels of high concentration samples were extracted with the corresponding target variables (i.e., $PM_{2.5}$ and $PM_{10}$) that were randomly perturbed within 5% of the focus pixel concentrations. This oversampling approach can effectively reduce the underestimation of high PM concentrations that results from the small training sample size of high concentration data."*

5 ➔ The adjusted dataset means a more balanced dataset through oversampling (and subsampling for a certain group) for effective training of a model, which results in improving the underestimation of high PM concentrations (refer to Figure 3).

3. Supp figure 2 is hard to read, and dots are just forming a black blanket…please improve.

➔ We replaced it with the heat-scatter plot to improve the readability.

[Figure]

*"Supplementary Figure 2: Comparison of PM concentrations to MODIS-derived AOD (left column) and GOCI-derived AOD (right column). 
[revised manuscript text omitted]